# Connexin 32 constrains a mesenchymal-like switch in differentiated urothelium and luminal cancers

Jennifer Hinley[1] , Simon C Baker[1] , Andrew S Mason[1] , Grigorios Kyriazis[2] , Omar Masood[3], Jennifer Southgate[1] 

The molecular programming of epithelial wound repair provides the origin for the signalling pathways that drive the growth and spread of carcinoma cells. Urothelium is the mitotically quiescent, barrier-forming transitional epithelium of the urinary tract, characterised by uniquely specialised superficial cells and a remarkable regenerative capacity in response to damage. Connexin 32 (Cx32) was expressed by differentiated human urothelium where it predominantly localised to the basolateral borders of superficial urothelial cells. Suppression of Cx32 gap junction intercellular communication did not affect differentiation but instigated the switch to a highly migratory, wound healing phenotype marked by TGF$\beta$-SMAD signalling, ECM-remodelling, and induction of mesenchymal and cell-cycle markers. Immunohistological classification of muscle-invasive bladder cancers revealed Cx32 expression to be informative in luminal tumour biology, with non-membrane localised Cx32 defining a Ki67-high, vimentin-expressing, and TGF$\beta$-activated subset of luminal tumours. Our findings identify Cx32 cell–cell communication as suppressing migratory and proliferative behaviours in normal urothelial differentiation and suggest that Cx32 assessment, within the context of luminal muscle-invasive bladder cancer, can predict more invasive biology. This reveals the potential for differentiated cancers to exhibit EMT.

## Introduction

Most bladder cancers arise by neoplastic transformation of urothelium, the transitional epithelium that lines the urinary tracts, including the bladder and ureters. Urothelium functions as a stable, low-turnover but self-repairing barrier evolved to protect the underlying tissues from exposure to pathogens and excreted urinary toxins. A dearth of targeted therapies for bladder cancer fuelled international interest in transcriptomic-based classification studies. By consensus, this has led to muscle-invasive bladder cancers (MIBCs) being broadly classified into "basal" and "luminal" subtypes, based on pathways driven, respectively, by P63 or PPARγ

signalling (Choi et al, 2014; Dadhania et al, 2016; Kamoun et al, 2020). Basal cancers show loss of urothelial identity by reverting to a more primitive basal/squamous epithelial form, with corresponding changes in cytokeratin expression, such as KRT5, KRT6A, and KRT14 gain and reflected in therapeutically relevant biology, such as autocrine-stimulated EGFR-driven growth (Rebouissou et al, 2014). By contrast, luminal MIBCs retain a similar profile of gene expression to normal differentiated urothelium, including KRT20 and uroplakins, and underpinning transcriptional regulators GATA3 and FOXA1 (Choi et al, 2014), with mutations and amplifications in PPARG contributing significantly to this subtype (Biton et al, 2014; Rochel et al, 2019). While this suggests that luminal cancers retain dependency on urothelial differentiation programming, the underpinning biology is less clear.

Human urothelium has evolved features that make it an exceptional urinary barrier. It has long been recognised as the least permeable mammalian epithelial barrier (Hicks et al, 1974; Negrete et al, 1996), since attributed to the superficial membrane-embedded uroplakin plaques that restrict transcellular solute permeability (Liang et al, 2001) and the highly developed tight junctions that limit paracellular permeability between adjacent superficial cells (Varley et al, 2006; Smith et al, 2015). In habit, urothelium is recognised as a stable, low turnover epithelium that balances mitotic quiescence with an exceptional capacity for self-regeneration and barrier repair in response to insult (Walker, 1960; Hicks, 1975). As yet, knowledge of the molecular pathways that instruct urothelial regeneration remains incomplete.

As well as tight junctions and E-cadherin containing adherens junctions, complexes between epithelial cells contain gap junctions. Responsible for direct connections between adjoining cells, gap junctions are implicated in supporting cellular differentiation and tissue homeostasis through both communication-dependent and -independent means. Formed as hexameric complexes from the 21-member family of connexins (Cx), gap junctions exhibit differential permeability to ions, metabolites, and second messengers, such as $Ca^{2+}$, $IP_3$, prostaglandin $E_2$, and adenosine derivatives (Valiunas et al, 2018; Zhao et al, 2022).

To study the contribution of Cx to the biology of normal human urothelium, we have used an in vitro system where surgically

[1]Jack Birch Unit of Molecular Carcinogenesis, Department of Biology and York Biomedical Research Institute, University of York, York, UK  [2]Department of Urology, Mid Yorkshire NHS Trust, Pinderfields Hospital, Wakefield, UK  [3]Leeds Kidney Unit, St James's University Hospital, Leeds, UK

Correspondence: jenny.baker@york.ac.uk; j.southgate@york.ac.uk

derived normal human urothelial (NHU) cells are expanded in a low calcium (0.09 mM), serum-free medium as serially propagated finite cell lines. These cultures display EGFR-regulated autocrine growth, are contact-inhibited at confluence and display a non-transitional, squamous (KRT14+/KRT13−/KRT20−) phenotype (Varley et al, 2005; Rebouissou et al, 2014). We have previously described two robust procedures to induce physiological and differentiated features of human urothelium in vitro. The first, referred to here as "TZPD," combines PPARγ activation by troglitazone (TZ), with concurrent inhibition of EGF-receptor signalling using PD153035 (Varley et al, 2004a). These defined conditions result in monolayer cultures expressing features associated with urothelial differentiation, including uroplakins, ELF3, and KRT20, although they do not form a functional barrier (Varley et al, 2004a; Bock et al, 2014; Fishwick et al, 2017). In the second, referred to here as "ABSCa," transitional differentiation proceeds following supplementation of the medium with 5% adult bovine serum (ABS) and 2 mM calcium (Cross et al, 2005). Although the specific drivers of differentiation-induction are less well defined because of the use of serum, the resulting "biomimetic" cultures are multilayered and display features characteristic of native urothelium, including organisation into basal, intermediate, and superficial cells and forming a functional tight barrier, as assessed by transepithelial electrical resistance (TEER) (Cross et al, 2005; Rubenwolf et al, 2012; Baker et al, 2014; Wezel et al, 2014; Hustler et al, 2018).

To perform the first systematic study of Cx expression by urothelium, we compared transcriptomes of matched NHU cell cultures in undifferentiated (KRT14+ and uroplakin−) and differentiated (KRT13+ and uroplakin+) states. We identified GJB1 as the most differentially expressed Cx transcript and verified that the encoded Cx, Cx32, was expressed by differentiated urothelium in the bladder and ureter, both in situ and in vitro. Manipulation of Cx32 expression and channel function revealed a role in tissue regeneration, allied to specific changes in cell phenotype, including activation of TGFβ pathway signalling and synchronous epithelial-mesenchymal plasticity. Studies of Cx32 expression in MIBC not only distinguished luminal Cx32+ from basal Cx32− subtypes but revealed insight into the biological differences underpinning variants of luminal MIBC. Here, we identify a novel subset of luminal MIBC defining some 50% of luminal tumours, characterised by non-membrane localised Cx32 and by enhanced TGFβ-signalling and Ki67 activity, a potential candidate subset for TGFβR-targeted therapy.

# Results

### Cx expression changes associated with urothelial differentiation

mRNAseq transcriptomes generated from cultures of finite (non-immortal) ureteric-derived NHU cell lines grown under low calcium (0.09 mM) serum-free growth conditions reflected a KRT14+ TP63+ squamous epithelial phenotype. Parallel cultures induced to differentiate as the result of either addition of serum and physiological calcium (ABSCa; [Cross et al, 2005]) or by pharmacological activation of PPARγ in EGFR-blocked cells using troglitazone and

PD153035 (TZPD; [Varley et al, 2004a]) were shown to possess a transitional phenotype. The transitional phenotype was confirmed by expression of epithelial cytokeratin transcripts including KRT13 and KRT20, barrier-forming tight junction CLDN3, CLDN4, and TJP3 transcripts; (Wu et al, 1994; Smith et al, 2015) and differentiation markers including urothelial-specific uroplakins, for example, UPK3A (Fig 1A).

The constitution of urothelial gap junctions (GJs) has not previously been described. Comparison of GJ transcript expression by NHU cell cultures in undifferentiated (squamous) versus differentiated (transitional) states, revealed significant twofold changes in Cx gene expression. GJA4 (Cx37) and GJB2 (Cx26) were associated with the KRT14+ squamous state, while GJB1 (Cx32) marked the KRT13+ transitional state (Fig 1B). Expression of five other Cxs; GJA1 (Cx43), GJA9 (Cx58/59), GJB7 (Cx25), GJC3 (Cx29), and GJD3 (Cx36), were also up-regulated in both in vitro differentiation models, but to a lesser extent (<twofold). From this, GJB1 (Cx32) was selected for further study.

### Influence of urothelial differentiation and PPARγ signalling on Cx32 gene and protein expression

To establish an experimental system for functional studies, the expression of GJB1 transcript and its encoded Cx32 protein was explored in vitro. Both differentiation protocols (ABSCa and TZPD) resulted in a marked increase in GJB1 expression when applied to either ureter or to bladder-derived NHU cells. These studies further demonstrated that manipulation of exogenous calcium alone to a concentration permissive for adherens and tight junction formation (Cross et al, 2005), was not sufficient to up-regulate GJB1 expression in the absence of differentiation-inducing ABS (Fig S1A).

The cellular localisation of Cx32 in vitro was assessed by indirect immunofluorescence. In control (non-differentiated) NHU cell cultures, some weak and diffuse cytoplasmic labelling was invariably observed. Irrespective of method (TZPD or ABSCa), differentiation resulted in an increase in Cx32 expression and de novo localisation to intercellular junctions within small areas where stratification was evident (Fig 1C).

Immunoblotting of NHU cell lysates revealed presence of the 32 kD monomeric form of Cx32 protein in all cultures and the relative amount of this species was unaffected by differentiation state. An overall induction of Cx32 protein expression was found following differentiation (either method), with the induced protein resolving to a band of ~54 kD, indicating the dimeric form of Cx32 (Fig 1D). Immunoblot detection of dimeric Cx32 is supported by reports of Cx32 in human, mouse and rat livers in which Cx32 resolved to monomeric and dimeric forms by SDS–PAGE because of the nature of connexon oligomerisation into hexameric channels (Green et al, 1988; VanSlyke & Musil, 2000; Nagy et al, 2003; Fowler et al, 2009).

We examined the role of PPARγ activation on Cx32 expression and localisation. Induction of GJB1 transcript (Fig S1B) and Cx32 protein (immunoblot Fig 1D and immunocytochemistry Fig S1C), was abolished by a specific and potent irreversible inhibitor of PPARγ, T0070907. In ABSCa-differentiated cultures, T0070907 reduced, but did not ablate, dimeric Cx32 expression,

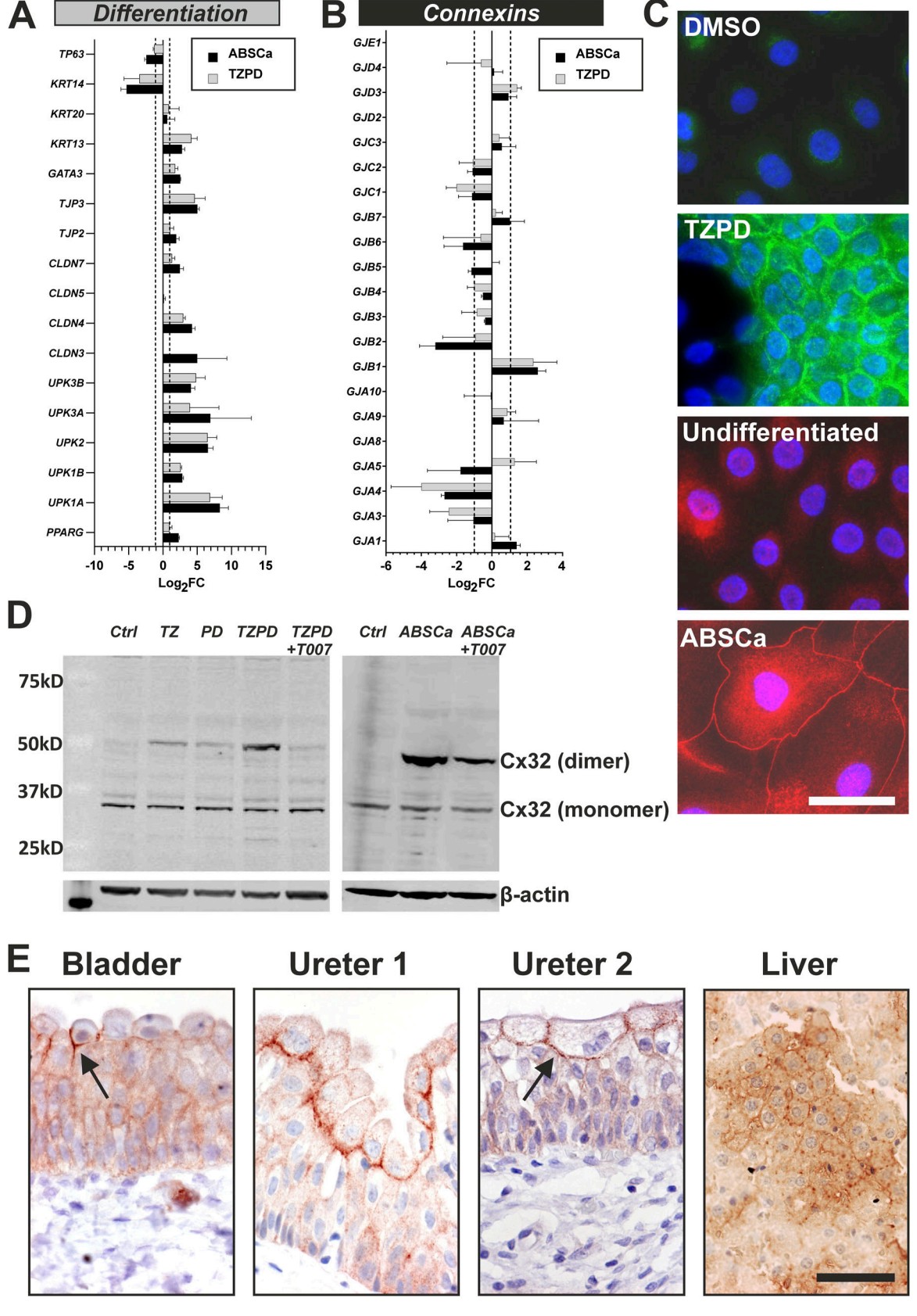

suggesting that additional drivers of Cx32 expression are present within serum (Fig 1D).

Given Cx can be rapidly turned over by the proteasome to regulate intercellular communication (Musil et al, 2000), we sought to understand whether differentiation-associated Cx32 induction was because of post-translational protein stability using the ubiquitination-proteasomal degradation inhibitor, MG132. In undifferentiated NHU cell cultures, we observed limited accumulation of dimeric Cx32 in the presence of MG132, but monomeric Cx32 protein was unaffected (Fig S1D). In differentiated (TZPD) cultures, MG132 treatment did not affect Cx32 expression, suggesting dimeric Cx32 protein in differentiated urothelium is stable and not rapidly turned over by degradation.

### Regulation of the *GJB1* gene in human urothelium

*GJB1* has two alternatively spliced tissue-specific promoters: P1 and P2. P1 is upstream of the non-coding exon 1a and drives *GJB1* expression in the liver and pancreas, while P2 lies upstream of the non-coding exon 1b and drives *GJB1* expression in neural tissue (Tomaselli et al, 2017). Analysis of mRNAseq data for differentiated NHU cells indicated that *GJB1* sequences mapped to exon 1a, with no reads mapping to exon 1b, thereby demonstrating that in human urothelial cells, *GJB1* expression is driven by the P1 promoter. Given the influence of PPARγ signalling on *GJB1* expression, we sought a potential PPAR response element within the *GJB1* P1 promoter; however, our promoter analysis did not predict a PPAR response element.

### In situ expression of Cx32

Immunohistochemical labelling of normal human bladder and ureteric urothelium, using two independent anti-Cx32 antibodies, demonstrated Cx32 to be most intensely concentrated along the basolateral membrane of superficial urothelial cells (Fig 1E). Weaker cytoplasmic and membrane localisation was observed in the underlying basal and intermediate cell layers.

### Generation of Cx32-modified urothelial sublines

In skin and mitotically quiescent liver, in vitro and in vivo studies revealed a complex association between Cx expression and the coordination of wound healing (Chanson et al, 2005; Becker et al, 2012; Pun et al, 2024). To explore the impact of Cx32 expression and channel function on urothelial regeneration, we developed stably transduced, NHU sublines to generate two experimental cell platforms: (a) Cx32 shRNA knock-down and (b) overexpression of WT or T134A-mutated Cx32. The T134A mutation has been applied to other Cx and targets a conserved residue which creates a dominant-negative closed gap junction channel, thereby blocking cell–cell communication. The T134A mutation does not interfere with trafficking or junctional assembly, enabling for identification of cellular behaviours that are communication-dependent, rather than expression/location dependent. The conserved threonine mutation eliminates communication in Cx26 and Cx43 homotypic and heteromeric pairings with WT protein, ensuring a dominant-negative effect (Beahm et al, 2006).

Immunoblotting confirmed both successful shRNA-knockdown of dimeric Cx32 protein and overexpression of Cx32$^{WT}$ and Cx32$^{T134A}$ (Fig 2A). By immunocytochemistry, plaques of Cx32 protein were observed at cell borders in both Cx32$^{WT}$ and Cx32$^{T134A}$ overexpressing NHU cultures (ABSCa differentiated; Fig 2B), indicating that overexpressed Cx32 was localised to the cell membranes regardless of wild-type or T134A-mutant status.

Scrape loading-dye transfer (SLDT), used to assess gap junction intercellular communication (GJIC; Fig 2C) (El-Fouly et al, 1987), was used to determine the functional effects of Cx32 shRNA-knockdown, or Cx32$^{WT}$/Cx32$^{T134A}$ overexpression, in transduced NHU cultures (differentiated with TZPD). SLDT confirmed significant loss of GJIC in Cx32 knock-down (Fig 2D) and Cx32$^{T134A}$ overexpressing differentiated NHU cultures, while overexpression of WT Cx32 showed significantly enhanced GJIC (Fig 2E). As an assay control, SLDT was applied in differentiated NHU cultures in the presence of the generic gap junction inhibitor 18α-glycyrrhetinic acid, which significantly disrupted dye transfer and, therefore, GJIC (Fig 2F).

### Cx32 suppresses migration of differentiated human urothelial cells in a communication-dependent manner

A scratch wound model was used to monitor the rate of wound repair in Cx32-modified NHU cultures differentiated with ABSCa. After scratch-wounding, time-lapse microscopy analysis revealed that Cx32 shRNA knock-down cultures were faster to migrate and

---

**Figure 1. Differentiation-associated expression of Cx32 (*GJB1*) by human urothelium.**
Gene expression changes by normal human urothelial cells after in vitro differentiation and analysis of RNAseq. **(A, B)** panel of archetypical urothelial differentiation-restricted genes for validation purposes and (B) connexin genes. The RNAseq data series represents three independent NHU cell lines differentiated in vitro using either ABSCa or TZPD protocols, for 7 and 6 d, respectively (Fishwick et al, 2017; Baker et al, 2020; Mason et al, 2022). Differentiation was confirmed from up-regulation of transcripts for the uroplakins (*UPK1A*, *UPK1B*, *UPK2*, *UPK3A*, and *UPK3B*) (Hu et al, 2002), tight junction-associated claudins (*CLDN3*, *CLDN4*, and *CLDN7*) (Varley et al, 2006), zonula occludens (*TJP2* and *TJP3*) (Rickard et al, 2008), *GATA3* (Fishwick et al, 2017), and transitional cytokeratins *KRT13* and *KRT20* (Varley et al, 2004b) as well as *PPARG*. Reciprocal loss was seen of squamous-associated *KRT14* and *TP63* (Fishwick et al, 2017). Data are represented as Log$_2$ fold-change TPM (transcripts per million) values, from differentiated NHU cell cultures relative to donor-matched undifferentiated parallel cultures. Error bars ± SD. Dotted line at ±1 indicates significant twofold change. **(C)** Cx32 immunofluorescence labelling of NHU cells cultured on glass slides for 6 d in vehicle control (DMSO) and TZPD differentiated, or undifferentiated and ABSCa differentiated, conditions. Nuclei were counterstained (blue) with Hoechst 33258. Scale bar = 25 μm. Red and green immunolabelling are the consequence of using secondary antibodies conjugated to different fluorochromes. **(D)** Left panel—Cx32 immunoblotting of NHU cells cultured for 6 d in 0.1% DMSO (vehicle control—Ctrl), 1 μM TZ, 1 μM PD153035, TZPD combined treatment, or TZPD with 5 μM T0070907. Right panel—Cx32 immunoblotting of NHU cells cultured in undifferentiated (Ctrl), ABSCa differentiated, or ABSCa with 5 μM T0070907. Monomeric and dimeric Cx32 forms are indicated and a β-actin loading control was included for each blot. **(E)** Immunohistochemical localisation of Cx32 on human bladder and ureter tissues, with rat liver Cx32-positive control tissue. Arrows indicate the focus of Cx32 gap junctions in the basolateral membrane of superficial cells. Scale bar = 25 μm.
Source data are available for this figure.

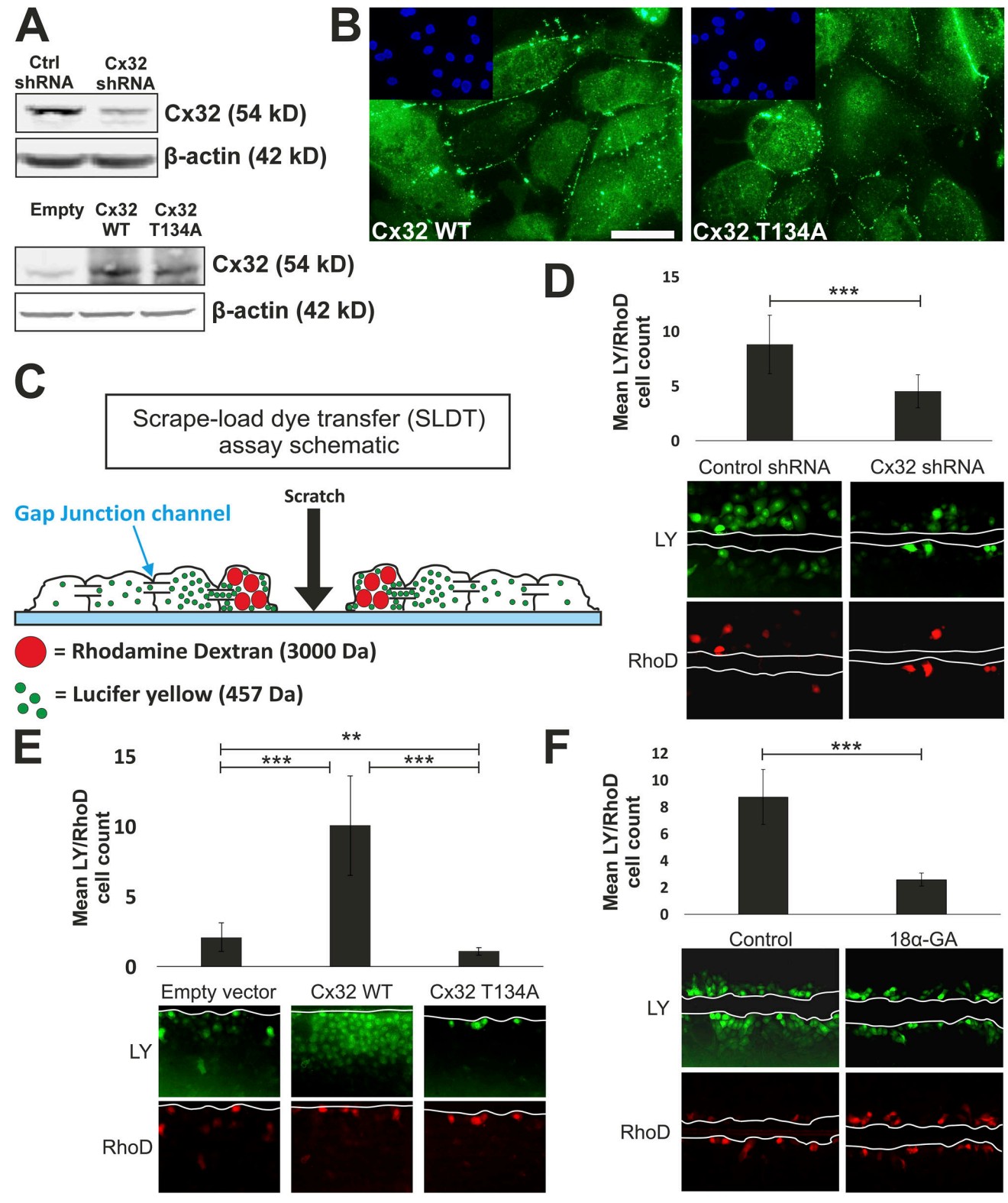

**Figure 2. Functional impact of Cx32-modification.**
**(A)** Cx32 protein (dimeric) expression detected by immunoblotting of ABSCa-differentiated cells following stable-transduction with control or Cx32 shRNA (upper panel), or overexpressing pLXSN vectors containing WT Cx32 (Cx32 WT), pore-closed mutant (Cx32 T134A) or empty vector control (lower panel). $\beta$-actin immunoblotting was included as a loading control. **(B)** Cx32 immunofluorescence labelling of ABSCa-differentiated NHU cells overexpressing WT Cx32 (Cx32 WT) or pore-closed mutant (Cx32 T134A). Nuclei were counterstained (blue) with Hoechst 33258 (insert). Scale bar = 25 $\mu$m. **(C)** Schematic to demonstrate the theory of scrape-load dye-transfer (SLDT). Cell monolayers are scraped in the presence of gap junction permeable (457 Da) lucifer yellow (LY; green) with rhodamine-dextran (RhoD; red, gap junction impermeable at 3,000 Da) to identify the initial dye-loaded cells. **(D, E, F)** SLDT to assess GJIC in NHU cultures (three independent NHU cell lines for (D), two independent

heal compared with control shRNA cultures (Fig 3A and B). In both cases migration was by collective and directional cell migration, or plithotaxis, with control shRNA cultures taking 22–24 h to fully heal (Fig 3B), compared with 12–14 h for Cx32 shRNA cultures. The $WC_{50}$ (mean time taken for the wound area to close by 50%) was significantly different between Cx32 knock-down and control cultures (7.7 h in Cx32 shRNA, versus 13 h in control; Fig 3C).

We sought to evaluate whether the suppressive effects of Cx32 on urothelial regeneration were dependent on GJIC. Time-lapse microscopy analysis was performed to study the rate of wound-healing in ABSCa-differentiated NHU cultures containing empty-vector control, or overexpressing $Cx32^{WT}$ and $Cx32^{T134A}$ proteins (Fig 3D). $Cx32^{T134A}$ pore-closed mutant cells were fastest to heal and $Cx32^{WT}$ cells the slowest to heal, with empty vector control cells healing at an intermediate rate. The $WC_{50}$ times from three independent NHU cell lines showed significant differences between empty vector, $Cx32^{WT}$, and $Cx32^{T134A}$ over-expressing cells (Fig 3E).

To evaluate whether the effects of Cx32 on urothelial regeneration were independent of GJIC, we assessed restoration of barrier function by measuring TEER after scratch wounding. Empty vector control, $Cx32^{WT}$, and $Cx32^{T134A}$ overexpressing cells were differentiated on permeable membranes with ABSCa, with each achieving a similar TEER during differentiation (2,500–2,900 $\Omega.cm^2$), suggesting that the channel status had no effect on barrier function. Mean TEER measurements reduced to <350 $\Omega.cm^2$ immediately after scratch-wounding (Fig 3F). TEER values in empty vector control, $Cx32^{WT}$, and $Cx32^{T134A}$ cultures remained low (<500 $\Omega.cm^2$) for 6 h post-scratching, with no significant difference observed between conditions. From 8 h post-scratching, barrier recovery was more rapid in pore-closed $Cx32^{T134A}$ cultures than empty-vector control cultures, while $Cx32^{WT}$ cultures displayed a much slower rate of barrier recovery. Significant differences in TEER were observed at 24-h post-scratch (Fig 3F).

Taken together, these findings indicate that intercellular signalling through Cx32 gap channels had no effect on barrier function but was inhibitory to repair and barrier restitution in differentiated urothelium.

### Transcriptomic analysis of Cx32-modified cells

Differential mRNAseq analysis was performed, comparing ABSCa differentiated NHU lines overexpressing $Cx32^{WT}$ and pore-closed $Cx32^{T134A}$ mutants. Our data revealed a significant (q < 0.05), >twofold induction of 50 genes in $Cx32^{T134A}$ cultures, with a further four genes significantly down-regulated relative to $Cx32^{WT}$ (Fig 4A and Table 1). No significant (>twofold) changes were observed of gene markers associated with basal/squamous or luminal MIBC biology (as reported by Choi et al [2014]) including

PPARG, GATA3, ELF3, and FOXA1, nor downstream differentiated target genes such as UPK3A, CLDN3, KRT13, or KRT20 (Fig S2A). We conclude that cultures over-expressing both $Cx32^{WT}$ and $Cx32^{T134A}$ retained transitional differentiation, with no gain or switch to squamous marker transcripts, such as KRT14 or KRT6A (Fig S2A).

Gene set enrichment analysis (GSEA) identified significant up-regulation of hallmark epithelial-mesenchymal transition "EMT" gene sets in $Cx32^{T134A}$ cultures (normalised enrichment score of 2.181, P-value 0.000 to 3 decimal places) (Fig 4B), and TGFβ-induced EMT datasets (normalised enrichment score of 2.163, P-value 0.000 to 3 decimal places), as reported in pan-cancer cell and tumour datasets (Foroutan et al, 2017) (Fig 4C). Specific interrogation of EMT-transcription factors (EMT-TFs) and key downstream target genes indicated acquisition of mesenchymal characteristics when Cx32 gap junctions were blocked (Fig 4D). We observed significant gains in mesenchymal marker genes, including vimentin (VIM), N-cadherin (CDH2), cadherin 10 (CDH10), and the transcription factor Slug (SNAI2). However, there was no loss or reduction in the epithelial marker E-cadherin (CDH1), nor up-regulation of other EMT-TFs (ZEB1/2, SNAI1, or TWIST1/2) or reporter transcripts fibronectin 1 (FN1) and P-cadherin (CDH3) (Fig 4D).

Because TGFβ pathway signalling is a prominent inducer of EMT and remodelling of the extracellular matrix, we sought to examine the effects of the $Cx32^{T134A}$ mutation on components of the TGFβ/BMP pathway and its downstream target genes. Among the transcripts significantly up-regulated in the $Cx32^{T134A}$ cells were TGFβ-superfamily ligands BMP1 and BMP2 (Fig S2B) and the BMP response genes ID1, ID2, ID3, ATOH8, GKN1, ANAX8, and SOCS2 (Fig 4E), which are transcriptionally regulated downstream of SMAD1/5 activation (Ramachandran et al, 2018). Other than BMP1 and BMP2, there were no other significant changes in TGFβ superfamily ligands, TGFβ/BMP receptors, or downstream SMAD family members (Fig S2 B–D). However, several TGFβ-SMAD3 target genes including ACKR3, COL12A1, CCDC80, LOX, EPHA4, MMP2, VCAN, and PTGS2 (Sethi et al, 2011; Brennan et al, 2012; Yeung et al, 2013; Castro et al, 2014; Cho et al, 2015; Wu et al, 2016; Hachim et al, 2017; Lian et al, 2021) were significantly up-regulated in the $Cx32^{T134A}$ cultures (Fig 4E). In $Cx32^{T134A}$ cultures, we also saw significant induction of key transcripts associated with tissue remodelling and ECM regeneration including CAV1, COL17A1, and LGALS1 (Fig S2E) and cell cycle entry such as CDK1, CDCA2, and RGCC (Fig S2F).

### The altered phenotype of Cx32-modified cells

A panel of antibodies was selected to verify the changes in TGFβ/EMT and cell cycle activity in differentiated (ABSCa) $Cx32^{WT}$ and $Cx32^{T134A}$ over-expressing NHU sublines from four donors. By immunoblotting, significant increases were observed in Slug,

NHU cell lines for (E, F)). TZPD-differentiated cultures were scrape-loaded in the presence of LY and RhoD fluorescent dyes. White lines mark the scratch boundaries on representative fluorescent images. **(D, E, F)** Bar chart indicates the mean number of LY positive cells, normalised to the number of RhoD loaded cells, from 10 (D, E) or 5 (F) contiguous areas across each scratch, to indicate the extent of dye-transfer capacity from a single "loaded" cell. Error bars = SD. Part (D) shows dye-transfer in control shRNA and Cx32 shRNA-transduced NHU cultures. ***P-value < 0.001 (unpaired t test). In (E), an assessment of gap junction-mediated dye-transfer was performed in pLXSN-empty, Cx32 WT and Cx32 T134A-transduced cultures. ***P-value < 0.0001, **P-value = 0.0057 (one-way ANOVA with Mann-Whitney two-tailed post-test). In (F), cultures were pre-treated for 30 min with the gap junction inhibitor 18α-glycyrrhetinic acid (18α-GA; 5 μM) or DMSO control. ***P-value < 0.001 (unpaired t test). Source data are available for this figure.

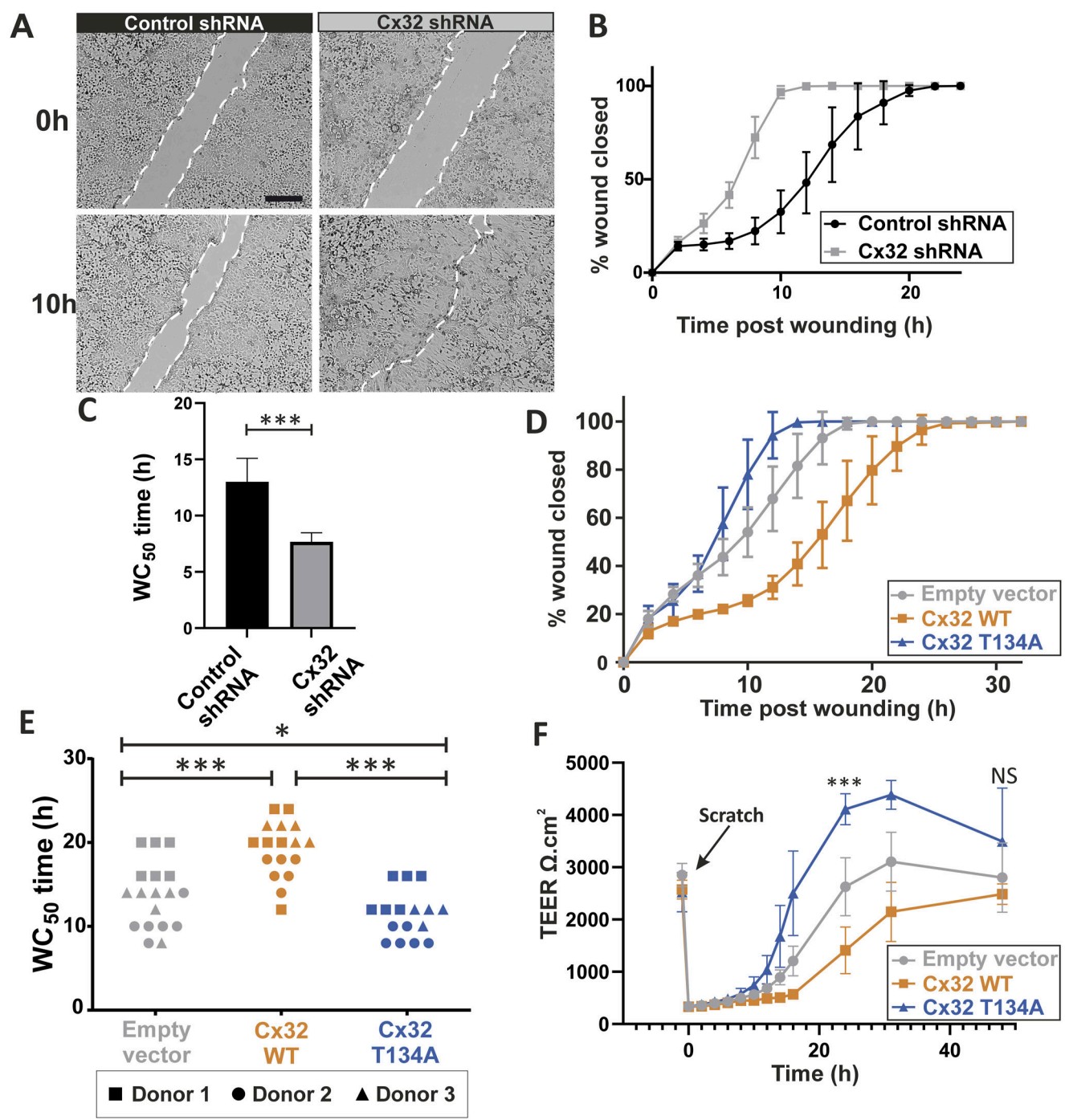

**Figure 3. Cx32 GJIC supresses repair of scratch-wounded differentiated human urothelial cell cultures.**
**(A)** Representative bright field images of ABSCa differentiated (7 d), control or Cx32 shRNA cells immediately and at 10 h post-scratch. A single scratch (~ 750 $\mu$m wide) was generated in cultures maintained in an environmental chamber and imaged every 2 h. Scale bar = 500 $\mu$m. **(B)** Wound healing analysis of control shRNA and Cx32 shRNA cultures differentiated with ABSCa. The mean percent closure of the wound area was plotted against time to demonstrate the rate of wound healing. Data represent six technical replicates from a single-transduced cell line. **(C)** The mean WC$_{50}$ (time taken for 50% wound closure) of six replicate control or Cx32 shRNA cultures. ***$P$-value <0.0001 (unpaired $t$ test). **(D)** Scratch repair analysis of control (empty vector), Cx32 WT, and Cx32 T134A overexpressing cultures differentiated with ABSCa. The mean percentage closure of the wound area was plotted against time to indicate the rate of repair. Data represent six technical replicates from a single-transduced cell line. **(E)** The WC$_{50}$ of control, Cx32 WT, and Cx32 T134A overexpressing sublines plotted for three independent donor cell lines (four to six replicate cultures each). ***$P$-value <0.0001, *$P$ = 0.026 (one-way ANOVA with Tukey's multiple comparisons post-test). **(F)** NHU cells stably transduced with pLXSN-empty vector, pLXSN-Cx32 WT, and pLXSN-Cx32 T134A, were differentiated with ABSCa on 0.4 $\mu$m permeable membranes for 6 d. A single scratch wound was generated (as indicated) in six replicate cultures for each group and TEER values were measured at intervals over a 48-h period. The mean TEER was plotted against time to demonstrate the rate of barrier repair. ***$P$ < 0.0001 for comparisons of the three transductants, NS, not significant (one-way ANOVA with Tukey's multiple comparisons post-test). All error bars are ± SD.

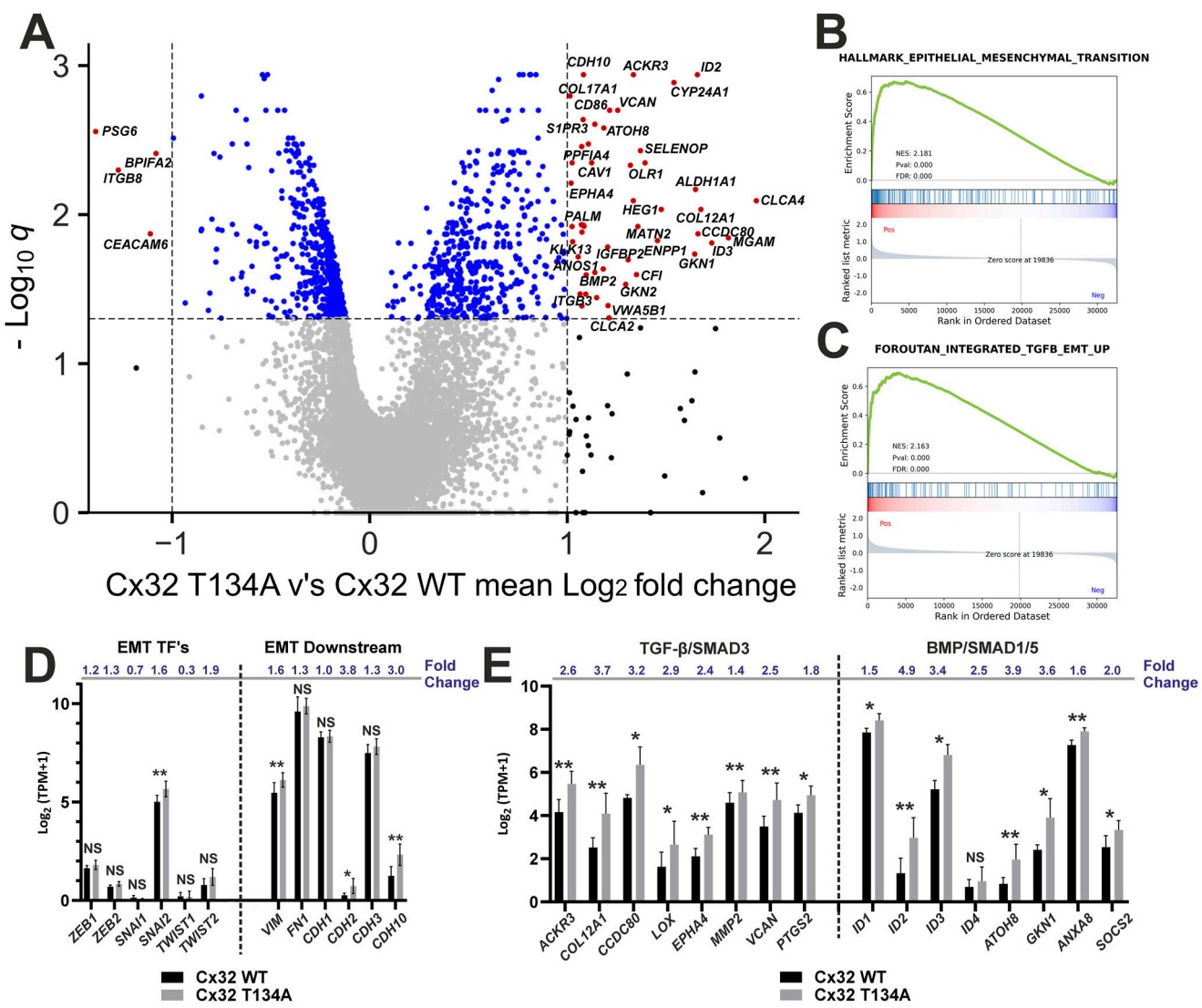

**Figure 4. mRNAseq analysis of differentiated (ABSCa) human urothelial cultures with modified Cx32 (*GJB1*) phenotype.**
**(A)** volcano plot demonstrating the significant (q < 0.05) >twofold-change increased gene expression by 50 genes in Cx32 T134A NHU cultures (red spots, top right panel) and down-regulation of four genes (red spots, top left panel), relative to Cx32 WT NHU cultures. Four independent NHU lines (two male and two female) were stably transduced to overexpress Cx32 WT or Cx32 T134A then differentiated in ABSCa for 6 d, before harvesting for mRNAseq. **(B, C)** Gene set enrichment analysis plots from the full mRNAseq dataset for Cx32 T134A NHU cultures, using donor-matched mean $\log_2$ fold change TPM to create a pre-ranked list of up/down regulated genes (relative to Cx32 WT). Plots for hallmark EMT and TGF$\beta$ datasets are shown. **(D, E)** RNAseq data expressed as transcripts per million (TPM, $\log_2$ transformed) for EMT-associated (D) or TGF$\beta$/BMP downstream target (E) genes, from Cx32 WT or Cx32 T134A differentiated (ABSCa) NHU cultures. * (q < 0.05), ** (q < 0.01), NS, not significant. Fold change values (Cx32 T134A relative to Cx32 WT) are indicated for each gene (mean values from four donors ± SD).

P-cadherin, phosphorylated SMAD3 and Cox-2 protein abundance in Cx32[T134A] cells (Figs 5A and S2G). Although not reaching statistical significance because of variance in the magnitude of response by cells from different donors, we also observed increased Vimentin and pSMAD1/5 in Cx32[T134A] cells from all four donors (Fig 5A). Expression of the epithelial markers E-Cadherin and Claudin 3 were similar between Cx32[WT] and Cx32[T134A] sublines from all four donors, as was expression of total SMAD3 protein, confirming that enhanced pSMAD3 abundance was because of increased TGF$\beta$ receptor signalling, rather than an increase in the available pool of SMAD3 protein (Fig 5A). The Cx32[T134A]

mutant is predicted by similarity to maintain normal protein: protein interactions (Beahm et al, 2006) and, therefore, over-expressing the Cx32WT protein on the same retroviral backbone is a relevant control for determining effects that are dependent on channel communication. Nonetheless, to mitigate any risk that our altered cell phenotype was caused by communication-independent effects, we verified protein abundance of key TGF$\beta$/EMT markers by immunoblotting of ABSCa-differentiated Cx32 shRNA NHU cells. Cx32 shRNA knock-down cultures showed significantly increased protein expression of vimentin and non-significant increases in pSMAD3 and pSMAD1/5, relative to

**Table 1.   Significantly up-regulated/down-regulated genes with >twofold change in differentiated NHU cells (ABSCa), transduced with overexpressing mutant Cx32$^{T134A}$ (ablated channel function) versus Cx32$^{WT}$ (WT).**

| Gene name | Gene title | q-value | Log$_2$ FC (Cx32$^{T134A}$ versus Cx32$^{WT}$) |
|---|---|---|---|
| CLCA4 | Chloride channel accessory 4 | 0.008 | 1.96 |
| MGAM | Maltase-glucoamylase | 0.014 | 1.82 |
| ID3 | Inhibitor of DNA binding 3 | 0.016 | 1.73 |
| COL12A1 | Collagen type XII alpha 1 chain | 0.009 | 1.68 |
| CCDC80 | Coiled-coil domain containing 80 | 0.013 | 1.66 |
| ID2 | Inhibitor of DNA binding 2 | 0.001 | 1.66 |
| ALDH1A1 | Aldehyde dehydrogenase 1 family member A1 | 0.007 | 1.65 |
| GKN1 | Gastrokine 1 | 0.018 | 1.65 |
| CYP24A1 | Cytochrome P450 family 24 subfamily A member 1 | 0.001 | 1.54 |
| NDUFA4L2 | NDUFA4 mitochondrial complex associated like 2 | 0.009 | 1.48 |
| ENPP1 | Ectonucleotide pyrophosphatase/phosphodiesterase 1 | 0.015 | 1.46 |
| OLR1 | Oxidized low density lipoprotein receptor 1 | 0.004 | 1.39 |
| SELENOP | Selenoprotein P | 0.004 | 1.37 |
| MATN2 | Matrilin 2 | 0.012 | 1.36 |
| SERPING1 | Serpin family G member 1 | 0.025 | 1.35 |
| ACKR3 | Atypical chemokine receptor 3 | 0.001 | 1.33 |
| HEG1 | Heart development protein with EGF like domains 1 | 0.008 | 1.33 |
| SCUBE2 | Signal peptide, CUB domain and EGF like domain containing 2 | 0.005 | 1.32 |
| CFI | Complement factor I | 0.020 | 1.31 |
| GKN2 | Gastrokine 2 | 0.029 | 1.29 |
| VCAN | Versican | 0.002 | 1.26 |
| CD200R1 | CD200 receptor 1 | 0.002 | 1.21 |
| CLCA2 | Chloride channel accessory 2 | 0.049 | 1.21 |
| VWA5B1 | Von Willebrand factor A domain containing 5B1 | 0.041 | 1.21 |
| IGFBP2 | Insulin like growth factor binding protein 2 | 0.017 | 1.20 |
| ATOH8 | Atonal BHLH transcription factor 8 | 0.003 | 1.18 |
| BMP2 | Bone morphogenetic protein 2 | 0.023 | 1.18 |
| LOX | Lysyl oxidase | 0.036 | 1.15 |
| IFITM10 | Interferon induced transmembrane protein 10 | 0.024 | 1.14 |
| CD86 | CD86 molecule | 0.002 | 1.14 |
| CAV1 | Caveolin 1 | 0.004 | 1.12 |
| PPFIA4 | PTPRF interacting protein alpha 4 | 0.003 | 1.11 |
| SUGCT | Succinyl-CoA:Glutarate-CoA transferase | 0.025 | 1.10 |
| LINC01835 | C-type lectin domain family 4 member O, pseudogene | 0.034 | 1.09 |
| PDLIM2 | PDZ and LIM domain 2 | 0.012 | 1.09 |
| ASPA | Aspartoacylase | 0.012 | 1.08 |
| CDH10 | Cadherin 10 | 0.001 | 1.08 |
| S1PR3 | Sphingosine-1-phosphate receptor 3 | 0.002 | 1.08 |
| NPIPA7 | Nuclear pore complex interacting protein family member A7 | 0.041 | 1.07 |
| GLIPR1 | GLI pathogenesis related 1 | 0.004 | 1.07 |

**Table 1. Continued**

| Gene name | Gene title | q-value | Log$_2$ FC (Cx32$^{T134A}$ versus Cx32$^{WT}$) |
|---|---|---|---|
| S100A3 | S100 calcium binding protein A3 | 0.013 | 1.07 |
| PALM | Paralemmin | 0.012 | 1.07 |
| ITGB3 | Integrin subunit beta 3 | 0.034 | 1.07 |
| ANOS1 | Anosmin 1 | 0.019 | 1.06 |
| KLK13 | Kallikrein-related peptidase 13 | 0.015 | 1.03 |
| LGALS1 | Galectin 1 | 0.004 | 1.02 |
| CHL1 | Cell adhesion molecule L1 like | 0.012 | 1.02 |
| EPHA4 | EPH receptor A4 | 0.006 | 1.02 |
| COL17A1 | Collagen type XVII alpha 1 chain | 0.002 | 1.01 |
| TRAV30 | T-cell receptor alpha variable 30 | 0.050 | 1.00 |
| BPIFA2 | BPI fold containing family A member 2 | 0.004 | −1.08 |
| CEACAM6 | CEA cell adhesion molecule 6 | 0.013 | −1.11 |
| ITGB8 | Integrin subunit beta 8 | 0.005 | −1.27 |
| PSG6 | Pregnancy-specific beta-1-glycoprotein 6 | 0.003 | −1.39 |

Fold change in transcripts per million reads was expressed as Log$_2$ fold change transformed data.

control shRNA cultures (Fig S2H and I), while E-cadherin was unaffected by the Cx32 shRNA knock-down.

Immunocytochemistry of differentiated ABSCa and Cx32$^{WT}$- and Cx32$^{T134A}$-transduced NHU sublines showed intense plaques of Cx32 protein at the cell membrane, as well in cytoplasmic vesicles, reflecting the abundance of overexpressed Cx32 protein (Fig 5B). Nuclear abundance of phosphorylated SMAD proteins was profoundly higher in Cx32$^{T134A}$ than Cx32$^{WT}$ cultures, indicative of enhanced TGF$\beta$ pathway signalling in Cx32$^{T134A}$ cells. Expression and localisation of total SMAD3 protein was unchanged between Cx32$^{WT}$ and Cx32$^{T134A}$ cultures.

The cell cycle status of Cx32$^{WT}$ and Cx32$^{T134A}$ overexpressing cultures, differentiated with ABSCa, was examined by immunofluorescence labelling for Ki67. Cx32$^{T134A}$ cultures displayed a significantly higher Ki67 labelling index than Cx32$^{WT}$ (Fig 5C). BrdU incorporation assay and propidium iodide cell cycle analysis was performed to determine whether the higher Ki67 index indicated active cell cycle progression through S-phase. No differences were found in either the number of cells incorporating BrdU (Fig 5D) or in the proportion of cells entering S or G2/M phases (Fig 5E). Together, these data suggest that the likely cause of increased Ki67 expression in Cx32$^{T134A}$ overexpressing cultures is that cells were growth-arrested in G1, rather than behaving like healthy differentiated urothelium where cells exit the cell cycle into G0 during periods of mitotic quiescence.

### Cx32 expression in luminal MIBC

As Cx32 GJIC acts to (a) suppress migration and mesenchymal transition and (b) promote cell-cycle exit into P0, in differentiated NHU cells, we sought to explore the expression and localisation of Cx32 in bladder cancer. A tumour microarray containing 118 tumour cores from a total of 55 patients with MIBC; age range 48–97 yr (median age = 72 yr), with a 1:4 female:male split was assessed by quantitative immunohistochemistry (IHC) for Cx32 and luminal identifiers PPARγ, GATA3, and FOXA1. Only tumour regions were analysed, with infiltrating lymphocytes excluded on the basis of nuclear size.

Quantification of Cx32 expression in the cell membrane and cytoplasm revealed three distinct localisation patterns. 44% (n = 24 patients) contained biopsies with Cx32 located at the cell membrane (with cytoplasmic labelling also present), a further 38% (n = 21 patients) contained Cx32 in the cytoplasm only (in the absence of membrane labelling), and 18% of patients (n = 10) had biopsies which were absent for Cx32 expression (Fig 6A and B). The Cx32 expressing tumours (membrane and cytoplasmic groups combined) were significantly enriched for the luminal MIBC markers GATA3 and FOXA1, but the association with PPARγ was non-significant (Fig 6C–E). Compared with the Cx32 membrane-expressing tumours, the Cx32-absent group was significantly enriched for CK5/6 expression, marking the basal/squamous histological subtype (Fig S3B).

Given that Cx32 protein expression in MIBC was more associated with luminal subtype markers, we performed a principal component analysis (PCA), based on gene expression profiles in The Cancer Genome Atlas (TCGA), the largest publicly available cohort of muscle-invasive bladder urothelial carcinomas. Our PCA was stratified into consensus molecular subtypes; basal/squamous (Ba/Sq), luminal papillary (LumP), luminal non-specified (LumNS), luminal unstable (LumU), stroma-rich and neuroendocrine-like (NE-like). As well as being the consensus framework within the bladder cancer field, our basis for electing to use the consensus classifier was to account for comprehensive basal/luminal markers, while gaining insight on the stromal contamination of the samples. We observed *GJB1* to cluster with the luminal genes *PPARG*,

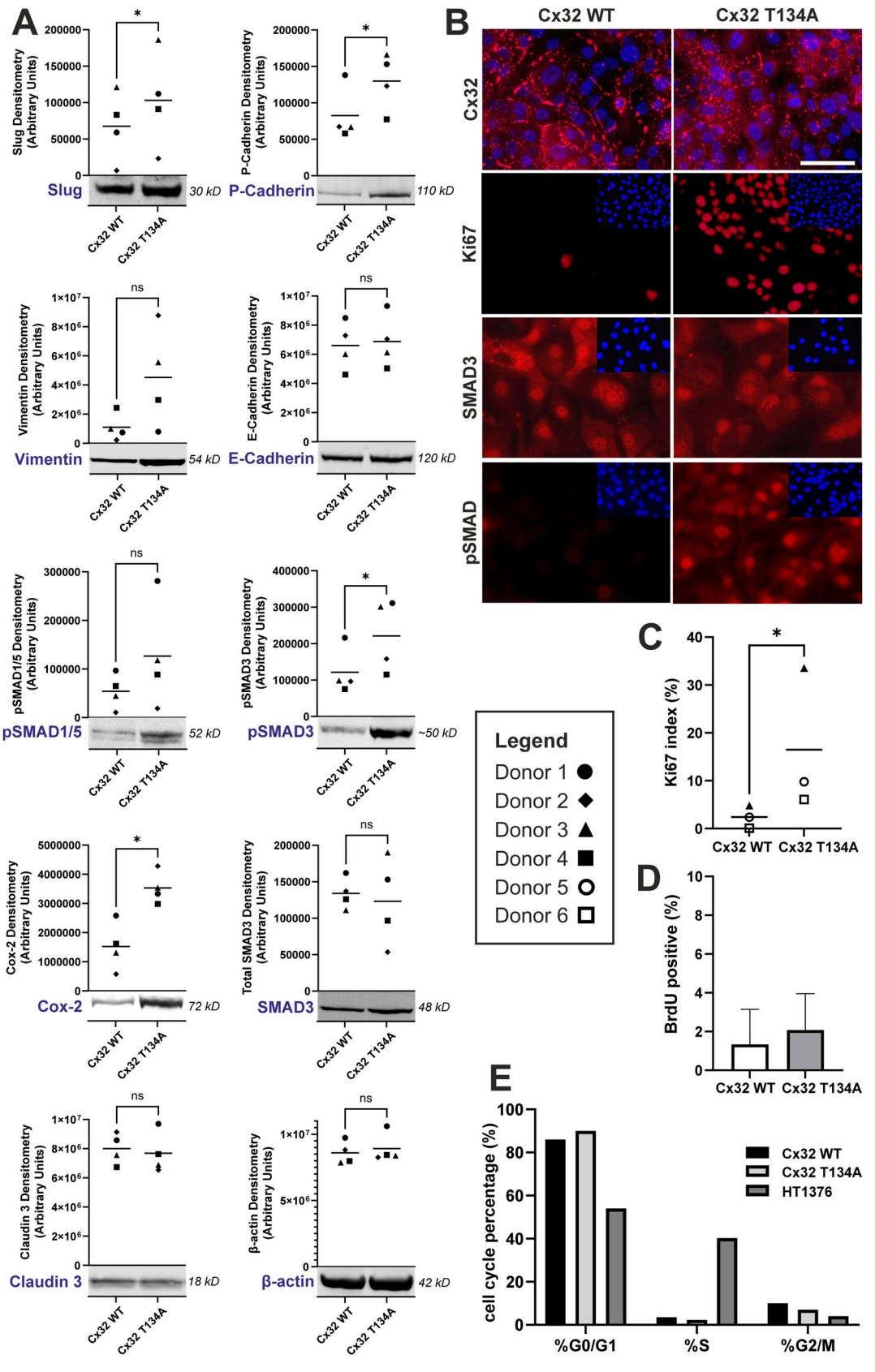

*GATA3, FOXA1, KRT20,* and *ELF3* but did not associate with expression of the basal/squamous genes *KRT5, KRT6A, KRT6B, KRT6C,* and *KRT14* (Fig S3A).

We next directed our attention to the Cx32-positive (luminal) tumours to establish whether Cx32 localisation to membrane or cytoplasmic sites was informative of the EMT-phenotype which was uncovered in our mRNAseq studies of differentiated Cx32^T134A over-expressing NHU cells. We performed quantitative analysis of Ki67, pSMAD (1/5 & 3), vimentin, and P-cadherin on our MIBC tissue microarray series (Fig 6F). The cell cycle marker Ki67 was significantly enriched in the Cx32-cytoplasmic group when compared with the Cx32-membrane tumours, as was pSMAD (1/5 & 3) and vimentin (Fig 6 G–J). We did not observe P-cadherin expression to be significantly different between the Cx32-membrane and Cx32-cytoplasmic groups (Fig S3B).

### Application of the Cx32^T134A regulon to luminal MIBC

In Cx32^T134A mutant differentiated NHU cells, 50 genes were significantly induced (>twofold). To evaluate whether this gene set was detectable in tumours, we used the TCGA publicly available bladder cancer dataset to evaluate co-regulation of these genes within bladder tumours, classified according to the consensus molecular classification (Kamoun et al, 2020). Because we have identified Cx32 to be a feature of luminal/differentiated biology, basal tumours were eliminated from our analysis. A Spearman rank correlation revealed a core group of 25 genes which significantly correlated within the TCGA bladder cancer dataset (Fig S4A). This Cx32 regulon was enriched in luminal NS (stroma rich) tumours (Fig S4B), raising the question whether these are infiltrating stromal cells or tumour cells undergoing EMT. Stratification of the luminal tumours into two groups using Euclidean distance to perform K means clustering, revealed a significant difference in survival between the two groups (Fig S4C), with outcomes significantly worse in patients who have high expression of this regulon.

## Discussion

We present the first systematic characterisation of Cx expression by urothelium, identify *GJB1* as urothelial differentiation associated and reveal a suppressive role for Cx32 in maintaining differentiated urothelial cells in a mitotically quiescent (G0) state. Supported by experimental verification from two human urothelial

in vitro differentiation platforms, our study reveals how blocking Cx32 intercellular communication promotes major changes in cellular phenotype and disruption of tissue homeostasis. While development of an integrated tight barrier proceeded normally from cells overexpressing the dominant negative-acting pore-closed Cx32^T134A mutant, there was a failure of the emergent tissue structure to exert a collective suppressive influence over individual urothelial cells, which displayed migratory, mesenchymal-like properties and failed to exit cell cycle. Nevertheless, it was notable that suppression of Cx32 GJIC in differentiated NHU cultures differed from the classical EMT switch (Yang et al, 2020), as while there was enhanced TGFβ pathway activity and gain of mesenchymal features such as vimentin and Slug, it was not accompanied by loss of differentiation features or ability to generate a functional tight barrier. This implies that ablation of Cx32 channel function results in a gain of mesenchymal markers as a result of cellular plasticity because of "escape" from the differentiated urothelial tissue framework.

We observed similar effects of enhanced TGFβ pathway activity and vimentin gain in Cx32 shRNA NHU cultures, allaying the risk of unanticipated off-target effects of the Cx32^T134A mutation on protein-protein interactions and cell signalling in NHU cells. Nonetheless, a complexity of connexons is their ability to form heteromeric channels with different connexons (Koval et al, 2014), including Cx32/Cx26 pairings. Cx26 is also expressed by urothelium, although not differentiation-associated like Cx32. A limitation of our study is the possibility that loss of Cx32 function may impact on the functionality of heteromeric connexons.

The localisation of membrane-bound Cx32 within the basolateral membrane of superficial cells indicates a role for communicating specific signalling molecules between superficial and intermediate cell zones. In HeLa overexpression cell lines, Cx26 gap junction coupling has been shown to limit the mitotic rate of the whole cell culture population, by redistributing cAMP throughout the cell population, thereby eliminating the cell-cycle oscillations in cAMP required for cell cycle progression (Chandrasekhar et al, 2013). One scenario is that shedding or damage to superficial cells removes Cx32-mediated suppression on juxtaposed cells that would otherwise inhibit efficient regeneration. Urothelium *in situ* may incur damage from a variety of sources, including uropathogens and urinary toxins. Damage or shedding of superficial cells has been observed as a key event in promoting regeneration of urothelium (Lavelle et al, 2002; Wiessner et al, 2022) and here we propose a unifying mechanism in which loss of superficial urothelial cells may eliminate the suppressive Cx32 channel effects on cell cycle state and migration.

---

**Figure 5. Influence of Cx32 GJIC on differentiated NHU cell phenotype.**
**(A)** Western blotting for slug, P-cadherin, vimentin, E-cadherin, pSMAD1/5, pSMAD3, SMAD3 (total), Cox-2, claudin 3, and β-actin (loading control) on Cx32 WT and Cx32 T134A-transduced NHU whole cell lysates following differentiation in ABSCa. Protein densitometry is plotted from four cell lines, as indicated in key. A representative blot from a single donor is shown, with all blots available in supplementary (Fig S2G). ns, not significant; *P < 0.05 (paired *t* test). **(B)** Immunocytochemistry for Cx32, Ki67, total SMAD3, and phospho-SMAD3 (ser 423/425; note antibody also detects pSMAD1/5 at equivalent phospho sites) on Cx32 WT and Cx32 T134A-transduced NHU sublines, differentiated in ABSCa. Nuclei were counterstained (blue) with Hoechst 33258. Scale bar = 50 μm. **(C)** Quantification of Ki67⁺ nuclei, expressed as a labelling index (%) in ABSCa-differentiated Cx32 WT and Cx32 T134A-transduced NHU cultures (counts from three fields of view, from three independent NHU donors as indicated in the legend) labelled by indirect immunofluorescence. Counts from three fields of view from n = 3 independent NHU lines (see the key); *P-value = 0.0479 (paired *t* test). **(D)** Quantification of 5-bromo-2′-deoxyuridine (BrdU) containing nuclei as a measure of S-phase activity in ABSCa-differentiated Cx32 WT and Cx32 T134A cells (16 replicate images from one donor). **(E)** Percentage of different cell cycle phases (G0/G1, G2/M, S) in ABSCa-differentiated Cx32 WT and Cx32 T134A cells assessed using propidium iodide. The HT1376 bladder cancer cell line was included as a highly proliferative positive control.
Source data are available for this figure.

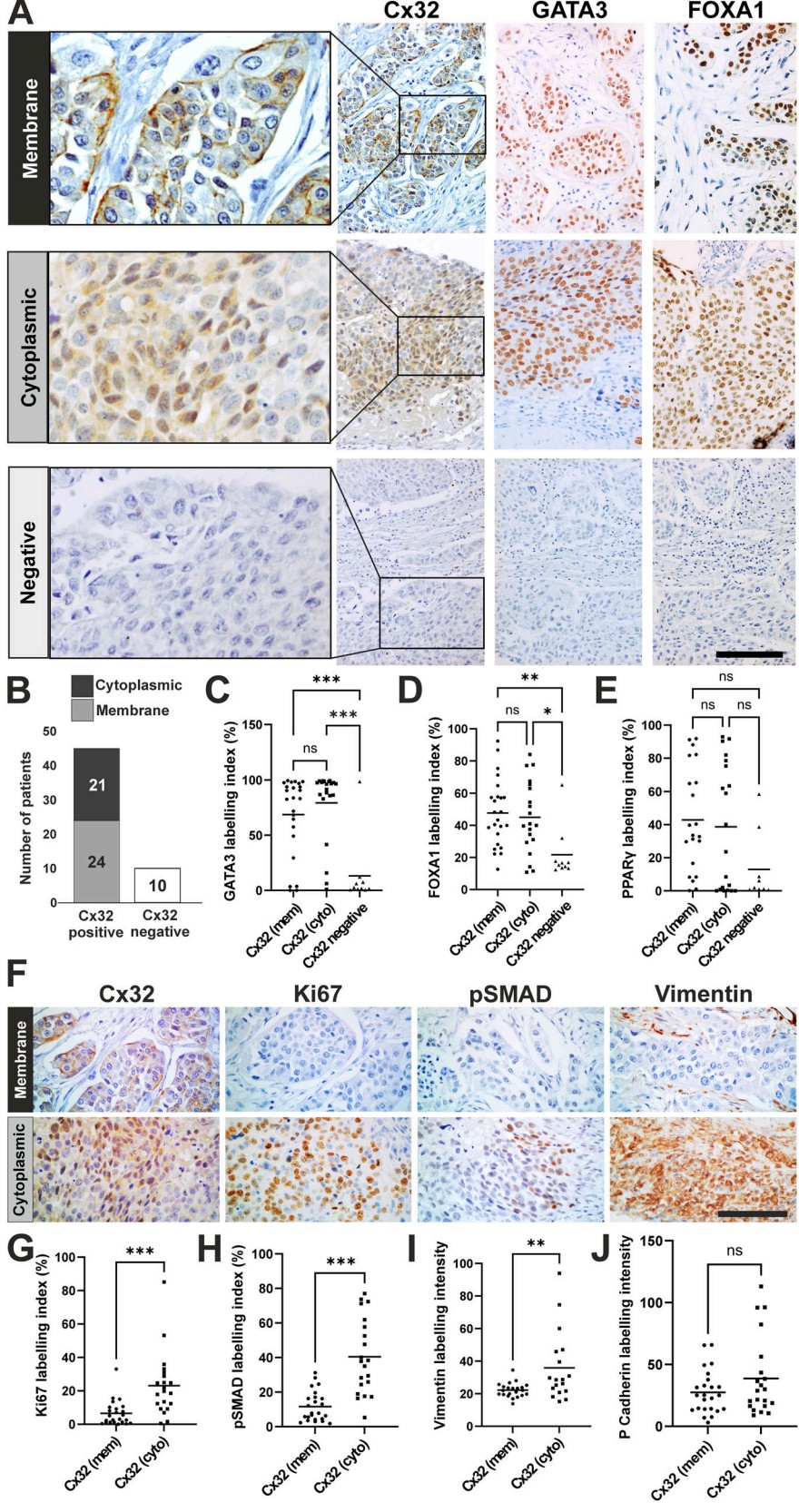

**Figure 6. Expression of Cx32 in muscle-invasive bladder cancer (MIBC).**
**(A)** Immunoperoxidase labelling of MIBC cores representing Cx32 localised to membrane (top panel) or cytoplasm (middle panel) or absent (bottom panel). Corresponding GATA3 and
FOXA1 immunohistochemistry for each biopsy is shown to distinguish luminal from basal-like cancers.
**(B)** Stratification of a cohort of 55 MIBC into Cx32-membrane, Cx32-cytoplasmic, or Cx32 absent groups, based on semi-automated image analysis (see the Materials and Methods section). **(C, D, E)** Immunoperoxidase labelling was quantified (labelling index) for GATA3, FOXA1, and PPARγ against Cx32-membrane (mem), Cx32-cytoplasmic (cyto) or Cx32-absent MIBC groups. *$P < 0.05$, **$P < 0.01$, ***$P < 0.001$, ns, not significant; one-way ANOVA with Tukey's post-test.
**(F)** Immunoperoxidase labelling of MIBC displaying either membrane or cytoplasmic Cx32 expression as indicated, with corresponding Ki67, pSMAD (pSMAD3 ser 423/425 and pSMAD1/5 at equivalent sites), and vimentin labelling. **(G, H, I, J)** Cx32-positive MIBC were assessed by intensity of immunoperoxidase labelling for Ki67, pSMAD (pSMAD3 ser 423/425 and pSMAD1/5 at equivalent sites), vimentin, and P-cadherin against membrane (mem) or cytoplasmic (cyto) Cx32 distribution. **$P < 0.01$, ***$P < 0.001$, ns, not significant; unpaired $t$ test. Scale bars = 100 $\mu$m. Note: the same tumour cores are used in (A and F) to represent membrane and cytoplasmic Cx32 localisations. Antibody labelling for each tumour was performed on serial sections and the images in each panel (running left to right) are captured from the same region within each tumour core.

This study provides a new perspective on how the urothelial differentiation programme offers a "luminal" path to malignancy, contrasting with the "basal/squamous" pathway that represents an escape from differentiation constraints. The major dichotomy between basal/squamous and luminal MIBC subtypes is defined by PPARγ (Choi et al, 2014; Dadhania et al, 2016). In NHU cells, PPARγ transactivates a cascade of intermediary transcription factors, including GATA3 (Varley et al, 2009; Fishwick et al, 2017), to promote development of an archetypical differentiated urothelial phenotype (Varley et al, 2004a, 2004b, 2006). Here, specific inhibition of PPARγ negatively impacted Cx32 gene/protein expression and eliminated membrane-associated Cx32, reinforcing its association with superficial urothelial cell biology. This, allied with our analysis of MIBC from the TCGA dataset, showed *GJB1* expression to be a feature confined to PPARγ-driven luminal MIBC subtypes. This luminal association was further confirmed by IHC, where expression of Cx32 correlated strongly with luminal-type MIBC and by localisation, further defined two subsets. An aberrant (non-membrane) localisation of Cx32 found in ~50% of luminal MIBC was associated with elevated Ki67 expression, activated TGFβ signalling and gain of mesenchymal-like characteristics. The relationship between PPARγ and Cx32 is not necessarily direct or causal, but their biology is intrinsically linked. Furthermore, the circuitry required for TGFβR pathway signalling is established within the PPARγ-driven urothelial differentiation programme (Fleming et al, 2012).

Tumour biology often equates malignancy with dedifferentiation and escape from the constraints of the specialised state. The molecular classification of MIBC into the two major subgroups of luminal and basal/squamous raises questions as to what selective growth or survival advantage exists for those tumours that retain dependency on the differentiation programme. An analogous role for Cx32 is described in the liver (Xu et al, 2024), another mitotically quiescent tissue renowned for its regenerative capacity. Analysis of rat livers after partial hepatectomy showed a temporary reduction in Cx32 mRNA and protein during regenerative repair (Kren et al, 1993), while aberrant expression of Cx32 protein is a feature of some hepatocellular carcinomas (Yang et al, 2017). Thus, unlike most epithelial tissues where differentiation commitment is a terminal state associated with the irreversible loss of proliferative capacity, epithelia such as urothelium and liver are held out of the cell cycle, but with cells primed to re-enter. In urothelium, this provides a path for cancers to exploit the luminal differentiated state. In the context of Cx32, one consideration is that cytoplasmic expression could support cellular invasion in a channel-independent role. Connexins possess significant channel-independent roles as reviewed in Wu and Wang (2019), with accumulation of cytoplasmic Cx32 correlated to invasion and metastasis of hepatocellular carcinoma cells (Li et al, 2007), although a clear mechanism for mislocalised Cxs in promoting metastatic behaviours remains to be elucidated.

The classification of MIBC into basal and luminal subtypes based on gene expression is an informative starting point for stratifying patients for disease management. However, for clinical application, it is necessary to capture the wider heterogeneity and biological profiles of bladder tumours. While we observed that Cx32 (*GJB1*) gene expression associates with PPARγ-driven luminal MIBC subtypes, we here show how subcellular location of Cx32 is highly informative of luminal tumour biology, which supports our experimental findings that Cx32 curbs urothelial invasive properties. This exemplifies the benefit of immunopathological profiling of tumours to gain information that cannot be resolved by transcriptomic profiling alone.

It has been consistently shown that activation of the TGFβ pathway is associated with poor survival and with resistance to immune checkpoint blockades caused by immune cell exclusion within TGFβ-rich tumour microenvironments (Mariathasan et al, 2018; Liu et al, 2021). We gave careful consideration to the analysis of MIBC molecular profiles in the context of TGFβ-driven EMT and the significant challenge in distinguishing mesenchymal biology of epithelial cancer origin, from the influence of a dominant stromal signature in low tumour purity biopsies, as discussed by Kreis et al (2024). Several molecular classification schemes have been proposed based on transcriptomic profiling of bladder cancers, including those by MD Anderson Cancer Centre (Choi et al, 2014), Institut Curie Cartes d'Identité des Tumeurs (CIT) (Rebouissou et al, 2014) and Lund University (Marzouka et al, 2018) and all sub-divide tumours based on "basal" versus "luminal" biology. The accuracy of the conclusions we draw in the present study benefit from the international effort to reach a consensus in MIBC subtypes (Kamoun et al, 2020), which accounts for stromal and immune infiltration in the samples. Even minor contributions from stromal cell transcriptomes prevented us from accurately classifying the epithelial component of tumours using our experimentally generated $Cx32^{T134A}$ transcriptomic signature to MIBC, without artefactual classification of tumours based on the presence of stroma. While our findings show poor clinical outcomes for patients where there is enrichment for our $Cx32^{T134A}$ regulon, we are cautious not to over interpret this because of the inherent "EMT-like" nature of the regulon and difficulty in distinguishing between (a) luminal tumour cells engaging in EMT-like behaviour or (b) infiltrating stroma. Nonetheless, it is relevant to note that this signature is associated with poor survival.

Connexins have been shown to supress transformed cell growth and invasion, though the underpinning tumour suppressive mechanisms are unclear. Here we present a detailed new insight into the role of Cx32-mediated GJIC in the homeostasis of normal urothelium, revealing how Cx32 GJIC enables cell cycle exit into G0. We demonstrate that loss of Cx32 GJIC primes cells for growth, promotes cellular migration and led to an altered phenotype with mesenchymal features. Our findings in normal urothelium have provided biological insight into mechanisms driving contrasting behaviours within luminal-type MIBCs and identify Cx32 localisation as a tool for further luminal MIBC stratification, with clinical relevance for selective therapy.

# Materials and Methods

### Tissue collection

This study was conducted and falls within the guidelines of the Declaration of Helsinki and the Department of Health and Human Services Belmont Report.

The collection and use of ureter tissue for research was approved by the Biology Ethics Committee for the University of York as JS202208_Study 99-095 and by the Leeds East National Health Service Research Ethics Committee (NHS REC) as project reference 99/095 (approval granted on 24th April 1999 and confirmed on 4th Sep 2024). Patient consent was waived as tissue was obtained as anonymous (unlinked) discarded tissue from transplant surgery and was used in the experimental studies reported here, including generation of finite NHU cell lines and IHC studies. In addition, bladder, ureter, and renal pelvis surgical samples were accessed from URoBank (NHS REC approved research tissue bank 16/YH/0396), containing anonymised donor specimens collected with informed patient consent. Use of MIBC specimens for research was approved in Germany by the appropriate institutional ethics committee (University of Regensburg; IRB number 08/108) and tissue microarray sections were transferred to the University of York under a Materials Transfer Agreement.

## Urothelial cell culture

NHU cells were isolated from surgical specimens and used to establish finite cell lines in keratinocyte serum-free medium supplemented with human recombinant epidermal growth factor and bovine pituitary extract at the time of use (KSFMc; Thermo Fisher Scientific), as described elsewhere (Southgate et al, 1994, 2002). For the experiments reported here, cell lines from n = 29 ureters, n = 1 bladder, and n = 1 renal pelvis derivation were used between passages 2–5.

Two previously reported protocols were used to induce differentiation of NHU cell cultures, involving either (a) combination use of the PPARγ agonist troglitazone (TZ, Tocris Bioscience at 1 $\mu$M) and the EGFR inhibitor PD153035 (PD, Calbiochem at 1 $\mu$M), referred to as "TZPD" (Varley et al, 2004a) or (b) 5% ABS (Seralab) supplemented to 2 mM CaCl$_2$, referred to as "ABSCa" (Cross et al, 2005). Control ("undifferentiated") cultures were maintained in KSFMc in parallel and harvested at matched time points as controls, with 0.1% (vol/vol) DMSO as solvent control where appropriate. Differentiation treatments were performed for 6 (TZPD) or 7 d (ABSCa) unless stated otherwise.

The PPARγ antagonist T0070907 (T007; Cambridge Bioscience Ltd.) was used at 5 $\mu$M by pre-treatment for 3–4 h before induction of differentiation. The proteasome inhibitor MG132 (Cayman Chemical) was used at 12.5 $\mu$M and the gap junction inhibitor 18$\alpha$-glycyrrhetinic acid (18$\alpha$-GA; Merck) at 5 $\mu$M.

## Generation of Cx32 (GJB1) modified cell lines

Cx32 shRNA knock-down, Cx32$^{WT}$ overexpressing, and Cx32$^{T134A}$ mutant (dominant negative) overexpressing sublines were developed from NHU cells isolated from three independent donors (Cx32 shRNA studies), or nine independent donors (Cx32 over-expression studies), using retroviral gene transfer technology under antibiotic selection.

### GJB1 gene silencing

Three different shRNA oligonucleotides were designed to target the GJB1 coding sequence (ENSG00000169562), with addition of a hairpin loop, restriction overhangs for directional cloning and a Mlu1 restriction site to verify cloned inserts, generating the following sense shRNA sequences:

GJB1 #1: gatccAATGAGGCAGGATGAACTGGATTCAAGA-GATCCAGTTCATCCTGCCTCATTTTTTTTACGCGTg.

GJB1 #2: gatccAACTGGACAGGTTTGTACACCTTCAAGAGAGGTGTA-CAAACCTGTCCAGTTTTTTTTACGCGTg.

GJB1 #3: gatccGCCGGCATTCTACTGCCATTTTCAAGA-GAAATGGCAGTAGAATGCCGGTTTTTTACGCGTg.

shRNA oligonucleotides were ligated into pSIREN-RetroQ vector (Clontech). After bacterial transformation, successful ligation was confirmed by Mlu1 restriction digest of purified plasmid. Plasmids were designated shRNA Cx32 #1, shRNA Cx32 #2, shRNA Cx32 #3 and shRNA control. A firefly luciferase non-targeting control shRNA was included (Clontech).

### GJB1 over-expression

Full-length GJB1 cDNA (WT) was amplified from RNA harvested from ABSCa differentiated NHU cells using the Expand High Fidelity PCR system (Roche) using oligonucleotides 5'-gaa ttc acc ATG AAC TGG ACA GGT TTG TAC ACC-3' (sense) and 5'-gga tcc TCA GCA GGC CGA GCA GCG GTC-3' (antisense). Primers incorporated a Kozac sequence (under-lined, sense), and restriction sites for EcoR1 (lowercase, sense) and BamH1 (lowercase, antisense). Amplicon was purified using the QIAquick Gel Extraction Kit (QIAGEN) and directionally cloned into the pLXSN vector (Clontech). The dominant negative Cx32 (Cx32$^{T134A}$) was generated by site-directed mutagenesis of WT GJB1 plasmid DNA following the single-primer reactions in parallel (SPRINP) method (Edelheit et al, 2009). The desired single base-pair mutation (T134A) was generated using the following primers (amino acid change underlined):

5'-GGACACTGTGGTG<u>GGC</u>CTATGTCATCAGC-3'(sense), 5'-GCTGAT GACATA<u>GGC</u>CCACCACAGTGTCC-3' (antisense).

After bacterial transformation, purified plasmid DNA was sequenced to verify successful cloning of full length Cx32 WT or T134A.

## Generation of stable sublines

pSIREN-RetroQ and pLXSN vectors, including empty vector controls, were transfected into PT67 packaging cells, from which virus-containing medium was harvested for retroviral transduction of proliferating NHU cultures, as previously described (Shaw et al, 2005). Transduced NHU cultures were subjected to antibiotic selection, either 1 $\mu$g/ml puromycin (Merck) for pSIREN-RetroQ or 0.1 mg/ml G418 (Merck) for pLXSN, with selection confirmed as successful by use of mock-transduced cultures. Following selection, transduced sublines were maintained under selection pressure using a maintenance dose of antibiotic (0.25 $\mu$g/ml for puromycin and 0.025 mg/ml for G418).

Overexpression and knock-down of Cx32 protein was verified by immunoblotting. Cx32 shRNA #1 gave the most effective knock-down of Cx32 protein and was, therefore, selected for use in subsequent shRNA experiments, unless otherwise stated.

 **Life Science Alliance**

## mRNA analysis

Total RNA was scrape harvested from cell cultures collected into TRIzol Reagent (Thermo Fisher Scientific) and cDNA was synthesised from 1 μg total RNA using the Superscript first-strand synthesis system (Thermo Fisher Scientific) according to manufacturer's instructions. DNase treatment was performed using the DNA-free kit (Ambion).

qRT-PCR was performed using primers designed with Primer 3 software and the SYBR Green system was followed according to manufacturer's instructions (Applied Biosystems). The following primer sets were used: *GJB1* (fwd): 5′-TTTGTACACCTTGCTCAGTGG-3′; *GJB1* (rev): 5′-GGAGATGGGGAAGAATTGGT-3′; *GAPDH* (fwd): 5′-CAA GGTCATCCATGACAACTTTG-3′; *GAPDH* (rev): 5′-GGGCCATCCACAGTC TTCTG-3′. Reactions (technical triplicates) were monitored using an ABI Prism 7900HT Sequence Detection System. Controls included RT-negatives for each cDNA sample, a non-template (cDNA negative) control and a genomic DNA positive control. *GJB1* gene expression was normalised to the house-keeping gene *GAPDH*.

For mRNA sequencing (mRNAseq), samples were submitted to the Oxford Genomics Centre (Oxford, UK) for Illumina HiSeq4000 mRNA sequencing with 75 bp paired end reads. Libraries were generated by standard Illumina polyA-selection protocols, using an average fragment length of 350 bp. Sequencing data quality was assessed with FastQC v0.11.5 (Andrews, 2012) and no reads were removed or trimmed. Reads were pseudoaligned to the Gencode v35 comprehensive annotation transcript FASTA file using kallisto v0.43.1 (Bray et al, 2016) and aggregated by gene using tximport v1.8.0 (Soneson et al, 2015) and biomaRt v2.37.0 (Durinck et al, 2009) in R v3.4.4. Differential expression analysis was conducted using the Sleuth v0.29.0 R package (Pimentel et al, 2017), accounting for matched donor genetic backgrounds between treatment groups, by implementation of the likelihood ratio test, generating Benjamini-Hochberg corrected *q*-values. RNAseq generated within this study can be found on the NCBI short read archive (accession PRJNA1220661). Fold change values used a TPM+1 transformation to reduce the influence of low abundance transcripts.

GSEA (Subramanian et al, 2005) was performed using *π*-values (Xiao et al, 2014), derived from the Sleuth likelihood ratio test q-values:

$$\pi = log_2\, fold\ change\ (TPM + 1) \times -log_{10}\ LRT_q.$$

GSEA was run using the pre-ranked list feature implemented in the python package GSEApy (0.10.2; available at https://github.com/zqfang/GSEApy). All genes with a *π*-value of zero were removed from the pre-ranked list as unranked data. The ranked list of genes was run against the Molecular Signatures Database (MSigDB) collection (msigdb.v7.5.1.symbols.gmt).

To identify transcripts associated with urothelial differentiation, previously generated mRNAseq datasets were used, comprising donor-matched, undifferentiated (PRJNA847878) and in vitro differentiated NHU cultures with either ABSCa (7 d; PRJNA610264) or TZPD (6 d; PRJEB14585) from three independent donors, as described previously (Fishwick et al, 2017; Baker et al, 2020; Mason et al, 2022).

## Promoter analysis

AliBaba2.1 was used to predict transcription factor binding sites in the *GJB1* promoter 1 sequence from the TRANSFAC 4.0 database (Grabe, 2002).

## Antibodies

The following primary antibodies were used for immunoblotting, immunofluorescence and IHC analyses: mouse monoclonal anti-Cx32 (clone 5F9A9; Thermo Fisher Scientific), rabbit polyclonal anti-Cx32 (#Ab66613; Abcam), rabbit polyclonal anti-Cx32 (#BS3527; Bioworld), mouse monoclonal anti-Ki67 (clone MM1; Leica Biosystems), rabbit monoclonal anti-SMAD3 (clone EP568Y; Abcam), rabbit monoclonal anti-pSMAD3 (ser423/425, also detects pSMADs 1/2/5 at equivalent phospho sites, clone EP823Y, Abcam), rabbit monoclonal anti-Slug (clone C19G7; Cell Signaling Technology), mouse monoclonal anti-vimentin (clone V9; Thermo Fisher Scientific), mouse monoclonal anti-E-cadherin (clone HECD-1; Abcam), mouse monoclonal anti-P-cadherin (clone OTI2D5; Biotechne), mouse monoclonal anti-*β*-actin (clone AC-15; Merck), rabbit monoclonal anti-Cox2 (clone D5H5; Cell Signaling Technology), rabbit polyclonal anti-Claudin 3 (#34-1700; Thermo Fisher Scientific), mouse monoclonal anti-claudin 4 (clone 3E2C1; Thermo Fisher Scientific), rabbit monoclonal anti-GATA3 (clone D13C9; Cell Signaling Technology), mouse monoclonal anti-FOXA1/HNF-3*α* (clone Q6; Santa Cruz Biotechnology), mouse monoclonal anti-cytokeratin 5/6 (clone D5/16 B4; Dako), mouse monoclonal anti-BrdU (clone 2B1; Enzo life Sciences).

## Immunoblotting

Cell cultures were scraped into SDS lysis buffer under reducing conditions containing 1 x protease inhibitors (cocktail set III; Merck). The protein concentration of lysates was measured using a Coomassie protein assay kit (Pierce). 20 μg of protein was separated by SDS–PAGE in MOPS or MES running buffer on 4–12% Bis-Tris NuPAGE gels (Thermo Fisher Scientific). On each gel a Precision Plus Protein ladder (Dual colour or All Blue; Bio-Rad) was loaded to support molecular weight estimation. Protein lysates were electrotransferred onto PVDF transfer membranes (Millipore), before blocking in Odyssey blocking buffer (LI-COR systems) or 5% non-fat dried milk for 1 h, as optimised. Membranes were probed with primary antibodies for 16 h at 4°C. Visualization was by incubation with secondary antibody, AlexaFluor 680 goat anti-rabbit or IRDye800-conjugated goat anti-mouse IgG (Thermo Fisher Scientific), before scanning and analysis on an Odyssey Sa infrared imaging system (LICORbio). Molecular weights were estimated by inclusion of Precision Plus Protein All Blue Standards (Bio-Rad). Full, uncropped blots were provided for review as source data.

## Indirect immunofluorescence

NHU cell cultures were grown and differentiated or treated (as required) on sterilised 12-well glass slides (C.A.Hendley Ltd, Essex), before fixation in either a 1:1 mixture of methanol:acetone or 10% (vol/vol) formalin. Where cells were formalin-fixed,

permeabilization was performed with 0.5% (wt/vol) Triton X-100 (Merck) in PBS. Primary antibodies were applied for 16 h at 4°C. After PBS washing and incubation with secondary antibody conjugated to Alexa 488 or Alexa 594, nuclei were visualised by staining with 0.1 $\mu$g/ml Hoechst 33258 (Merck). Cells were examined under epifluorescence using an Olympus BX60 microscope, with UPlanApo 20x (numerical aperture 0.8) and 60x (NA 1.4) lenses and imaged using a DP74 camera and cellSens software (Olympus).

### SLDT assay

To assess GJIC, a SLDT technique (El-Fouly et al, 1987) was adapted for NHU cultures as follows: confluent NHU cultures in 24-well plates were differentiated in TZPD (Varley et al, 2004a), for 6 d to induce Cx32 expression. These assays were not performed in NHU cultures differentiated with ABSCa because of challenges with imaging highly stratified cultures. Cultures were scratched with a 21 gauge needle in the presence of 1 mM lucifer yellow (LY; 457.25 Da) and 0.5 mg/ml rhodamine dextran (RhoD; 3 kD) (both Life Technologies) and incubated for 6 min at 37°C to permit dye transfer, before fixation in 10% (vol/vol) formalin and counterstaining with Hoechst 33258 (Merck). The fixed preparation was visualised by epifluorescence on an Olympus IX81 motorized inverted microscope equipped with UPlanFl 10× (NA 0.3) and 20× (NA 0.4) objectives. Ten consecutive fluorescent images along the scratch were analysed using TissueQuest (TissueGnostics). The "nuclear mask" function was used to distinguish cells based on the Hoechst 33258 stain. The software was then used to perform automated counts to determine the number of RhoD- and LY-positive cells, after manual setting of a background threshold for each channel. For each field of view the number of LY-positive cells was normalised to the number of RhoD cells and mean values were taken to be proportional to the GJIC capacity of the culture. Control cultures containing the potent gap junction inhibitor 18$\alpha$-glycyrrhetinic acid (18$\alpha$-GA, 5 $\mu$M) were included to demonstrate that dye-transfer was by means of functional gap junctions.

### Timelapse microscopy

NHU cells were grown to confluence in 24-well plates and differentiated with ABSCa. Cultures were scratched with a 200 $\mu$l pipette tip to generate a wound approximately 750 $\mu$m wide. Wound-repair was monitored in an environmental chamber set to 37°C (Solent Scientific) by phase-contrast timelapse microscopy, using an Olympus IX81 inverted microscope equipped with UPlanFl 10× (NA 0.3) and 20× (NA 0.4) lenses and motorized stage. Images were captured every 2 h for 36 h, using CellM image capture software (Olympus) and were analysed by measurement of the wound area expressed as the percentage healed compared with the original wound area, in replicate (n = 6) cultures.

### TEER

NHU cells were seeded onto 0.4 $\mu$m permeable ThinCert membranes (Greiner Bio-One), at 5 × 10$^5$ cells per 1.1 cm$^2$ membrane (three to six replicate cultures per experimental condition) and differentiated with ABSCa. TEER was measured using an EVOM

voltohmeter (World Precision Instruments) as a functional assessment of urothelial barrier properties, using >500 $\Omega$.cm$^2$ to define a tight barrier (Fromter & Diamond, 1972).

### BrdU assay

Changes in S phase activity following Cx32 modification (overexpression of Cx32$^{WT}$ or Cx32$^{T134A}$) was assessed in NHU cells differentiated using ABSCa on sterilised 12-well glass slides. The growth medium was supplemented with 0.03 mg/ml 5-Bromo-2'-deoxyuridine (BrdU) (Merck) for 24 h, before fixation in methanol: acetone (vol/vol). After hydrolysation in 1.5 N hydrochloric acid for 30 min at ambient temperature and washing in PBS, incorporated BrdU was detected with mouse anti-BrdU antibody by incubation at 4°C for 16 h. Bound antibody was visualised using ImmPRESS™ Excel polymer staining kit (Vector labs) according to manufacturer's instructions. 3'3-diaminobenzidine (DAB; Merck) was applied as chromagen solution, before counterstaining for 5 s in Mayer's haematoxylin (Merck). Four images were taken from four technical replicate cultures (16 images comprising of 2,600–3,000 cells). The number of positive and negative cells was counted to obtain a mean percentage of BrdU-positive cells (S-phase index).

### Cell cycle analysis

Changes in cell cycle activity following Cx32 modification (overexpression of Cx32$^{WT}$ or Cx32$^{T134A}$) were assessed in NHU cells differentiated using ABSCa on 0.4 $\mu$m permeable ThinCert membranes (Greiner Bio-One), seeded at 5 × 10$^5$ cells per 1.1 cm$^2$ membrane (six replicate cultures).

Cell sheets were detached from the ThinCert membranes by incubation in 8 U/ml dispase II (Merck) for 20–30 min at 37°C and a single cell suspension was harvested by incubation in 0.1% (wt/vol) EDTA in PBS for 5 min at 37°C followed by a further 5 min incubation at 37°C in 0.25% (wt/vol) trypsin versene solution in Hank's balanced salt solution (HBSS; Life Technologies) containing 0.02% (wt/vol) EDTA. After inactivating the trypsin in 5% serum, cells were collected by centrifugation (300$g$ for 2 min), fixed in ice-cold 70% ethanol for 30 min, washed in PBS, and labelled with 400 $\mu$g/ml PI and 1 mg/ml RNAse A (both Merck), diluted in PBS for 30 min at 37°C. Samples were analysed on a CyAn ADP flow cytometer (Beckman Coulter), excitation 488 nm laser, emission 613/20-area. Cells were gated from debris using forward versus side scatter and single cells gated from aggregates using 613/20-height versus 613/20-area. Data were analysed using FCS Express software (Dotmatics).

### IHC and semi-quantitative analysis

Five $\mu$m paraffin-wax embedded tissue sections were heated to 72°C for 16 h before dewaxing in xylene and rehydrating into water. Sections were treated to block endogenous peroxide and avidin/biotin sites and antigen retrieval was performed by autoclaving for 30 min at 95°C in 10 mM citric acid buffer, pH 6.0. Sections were incubated with primary antibodies at 4°C for 16 h. Bound Cx32, FOXA1, and Ki67 primary antibodies were visualised using

biotinylated secondary antibodies (DAKO) in combination with the Vectastain Elite ABC kit (Vector labs), used according to the manufacturer's instructions. Bound pSMAD, vimentin, PPARγ, P-cadherin, and GATA3 primary antibodies were visualised using ImmPRESS Excel polymer staining kit (Vector labs) used according to the manufacturer's instructions. 3'3-diaminobenzidine (DAB) was applied to all slides as the chromagen, before counterstaining in Mayer's haematoxylin. Cytokeratin 5/6 labelling was performed on the Leica Bond 3 platform using Epitope Retrieval Solution 2 (AR9640; Leica Biosystems) for 30 min, with primary antibody application of 15 min, the Bond Polymer Refine Detection Kit (DS9800; Leica Biosystems) and Bond DAB enhancer (AR9432; Leica Biosystems) for 5 min.

For analysis of TMAs, digitised slides were generated using an AxioScan.Z1 slide scanner (Zeiss). Regions of interest were manually drawn around all tumour areas to strictly limit contamination of the analysis with stromal cells and a minimum nuclear size threshold of 30 $\mu m^2$ was set to exclude infiltrating T-lymphocytes from the analysis. A section of normal human ureter was mounted onto each TMA slide as an internal experimental control.

For nuclear antigens, DAB expression was quantitatively analysed using HistoQuest software (v3.5; Tissue Gnostics), using an algorithm that detected nuclei based on the haematoxylin counterstain. For cytoplasmic antigens (vimentin and P-Cadherin), the DAB intensity was assessed after applying a ring mask. A working example of the vimentin "ring mask" analysis is shown in Fig S5. For Cx32, two analyses were performed. In the first, a "skeleton" algorithm was used to identify membrane labelling which was attributed to the adjacent nuclear haematoxylin counterstain to calculate an angle of staining per cell (expressed in in degrees). In the second analysis, cytoplasmic DAB intensity was quantified using a ring mask surrounding the algorithm-detected nuclei. Labelling thresholds were set at 7.5° for membrane Cx32 and 10 (arbitrary units) for Cx32 cytoplasmic ring mask intensity, below which samples were considered to be negative.

### PCA of transcriptomic data from TCGA

TCGA mRNAseq (Robertson et al, 2017) was converted to transcripts per million (TPM) and $\log_2$(TPM + 1) values were used for PCA. The published consensus subtypes (Kamoun et al, 2020) ("basal/squamous," "luminal papillary," "luminal non-specified," "luminal unstable," "neuroendocrine-like," and "stroma-rich") were used to colour the data-points. Gene expression was analysed in an unscaled principal component analysis using prcomp in R (v 3.6.1). Graphs were drawn by fviz pca (v 1.0.3) in R.

### Generation of the Cx32^T134A^-regulon

The list of 50 genes identified as significantly increased by >twofold ($q$ < 0.05) by Cx32^T134A^ in differentiated NHU cell cultures was extracted from the TCGA MIBC bladder cancer expression cohort (Robertson et al, 2017). Counts data were converted to TPM values. "Neuroendocrine-like" and "basal/squamous" tumours were excluded from our analysis using the published consensus subtypes (Kamoun et al, 2020) because Cx32 is a feature of luminal tumours. Because the "stroma-rich" class of the consensus molecular

classification describes urothelial purity of the sample (i.e., the presence of other cell types) and not urothelial differentiation as for the remaining five classes, the published centroids were used to reclassify the stroma-rich class as the best fit urothelial class (i.e., Ba/Sq, LumNS, LumP, LumU, and NE-like). Consensus "stroma-rich" tumours with a best fit urothelial class of "neuroendocrine-like" or "basal/squamous" were again excluded.

To refine the signature, Spearman correlation matrices were calculated to evaluate co-regulation of the gene set within luminal tumours, with pairwise median Spearman Rho correlations calculated for each gene. Genes with an average Spearman Rho value > 0.5 ($n$ = 25; gene list in Fig S4B) formed the transcriptomic "Cx32^T134A^ regulon," where all genes were >twofold increased in Cx32^T134A^ cultures in vitro and closely correlated in TCGA luminal tumours.

Initial evaluation of the Cx32^T134A^ regulon in tumours was performed on $\log_2$(TPM + 1) data using hierarchical clustering (based on Euclidean distance and K means clustering) to generate two groups (Fig S4), respectively. Clustering analysis and heatmap generation was performed using Morpheus (Broad Institute; https://software.broadinstitute.org/morpheus/). Kaplan-Meier plots were made using Prism v9 (Graphpad) including statistical analysis by Log-rank (Mantel-Cox) test ($P$ = 0.010) and Gehan-Breslow-Wilcoxon test ($P$ = 0.019) supporting statistically significant differences in survival and a Mantel-Haenszel hazard ratio of 1.77.

## Data Availability

Original sequencing data for the Cx32-engineered cell lines used within this study can be found on the NCBI short read archive (accession PRJNA1220661). Previously generated mRNAseq data-sets were used in this study, comprising donor-matched, undifferentiated (PRJNA847878), and in vitro differentiated NHU cultures with either ABSCa (7 d; PRJNA610264) or TZPD (6 d; PRJEB14585) from three independent donors, as described previously (Fishwick et al, 2017; Baker et al, 2020; Mason et al, 2022).

## Supplementary Information

## Acknowledgements

This study was funded on a programme grant awarded to J Southgate by York Against Cancer. SC Baker was supported by a Kidney Research UK fellowship (INT_006_20210728). We thank all our urology colleagues who supply tissue specimens for research, in particular, Peter Rubenwolf, Wolfgang Otto, and Maximilian Burger for supplying the Tissue Microarrays. We would also like to thank Lisa Kirkwood, Celina Whalley, Meg Stark, Karen Hogg, and Jo Marrison for technical assistance. We are grateful to Dr Jens Stahlschmidt for expertise with CK5/6 immunohistochemistry and to the team at TissueGnostics, in particular Rupert Ecker and Cosmin Popovici, for their technical support with immunohistochemistry quantitation. The

generation and initial processing of the sequencing data was performed at the Oxford Genomics Centre at the Wellcome Centre for Human Genetics.

## Author Contributions

J Hinley: conceptualization, data curation, formal analysis, validation, investigation, visualization, methodology, and writing—original draft, review, and editing.

SC Baker: data curation, formal analysis, validation, investigation, visualization, and writing—review and editing.

AS Mason: data curation, formal analysis, validation, investigation, and writing—original draft.

G Kyriazis: resources.

O Masood: resources.

J Southgate: conceptualization, resources, supervision, funding acquisition, validation, investigation, visualization, methodology, project administration, and writing—original draft, review, and editing.

## Conflict of Interest Statement

The authors declare that they have no conflict of interest.

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
