## [Reviewer comments · Life Science Alliance]

Connexin 32 constrains a mesenchymal-like switch in differentiated urothelium and luminal cancers

Jennifer Hinley, Simon Baker, Andrew Mason, Grigorios Kyriazis, Omar Masood, and Jennifer Southgate
DOI: <https://doi.org/10.26508/lsa.202503427>

Corresponding author(s): Jennifer Southgate, University of York and Jennifer Hinley, York Biomedical Research Institute

Review Timeline:

Submission Date:	2025-06-19
Editorial Decision:	2025-07-18
Revision Received:	2025-11-28
Editorial Decision:	2026-01-09
Revision Received:	2026-01-29
Accepted:	2026-01-30

Scientific Editor: Sarita Hebbar

Transaction Report:

July 18, 2025

Re: Life Science Alliance manuscript #LSA-2025-03427-T

Prof. Jennifer Southgate
University of York
Department of Biology (Area 13)
Jack Birch Unit
Heslington
York YO10 5DD
United Kingdom

Dear Dr. Southgate,

Thank you for submitting your manuscript entitled "Connexin 32 constrains a mesenchymal-like switch in differentiated urothelium and luminal cancers" to Life Science Alliance. The manuscript was assessed by three expert reviewers, whose comments are appended to this letter.

As you will see, the reviewers disagreed on the overall suitability of this work. We appreciate the concerns of Reviewer 1 however we concur with Reviewers 2 and 3 on the importance and novelty of these findings

Reviewer 1 has asked for additional information on the nature and representation of the different cell types present in the culture. We concur that this must be addressed in a manner of your choice. Reviewers 2 has stressed the need to have a more in-depth analyses of the RNAseq dataset to include basal/luminal, EMT markers, and for an additional comparison, between Cx32 knockdown vs control, to rule out gain of function effects. In this connection, Reviewer 3 suggested that the altered SMAD/TGF β signalling observed with T134A should be tested in Cx32 shRNAi. We agree that a revised manuscript must address these points. In the context of tumour related analyses, Reviewers 1 and 2 have asked for a more detailed description of Cx32's link to cancer via use of TCGA database. They have provided specific suggestions on reporting the correlation. We leave it to you to decide the additional parameters to report on based on their suggestions. Finally we agree with Reviewer 3 that the conclusion, that Cx32 is a biomarker, needs to be toned down or supported by causal evidence.

We invite you to submit a revised manuscript addressing the Reviewer comments. When submitting the revision, please include a letter addressing the reviewers' comments point by point. While a rebuttal must respond to all points in some form, additional experiments to resolve these points, other than indicated above, will not be required.

Thank you for this interesting contribution to Life Science Alliance. We are looking forward to receiving your revised manuscript.

Sincerely,

Sarita Hebbar, PhD
Scientific Editor
Life Science Alliance
<http://www.lsjournal.org>

B. MANUSCRIPT ORGANIZATION AND FORMATTING:

Reviewer #1 (Comments to the Authors (Required)):

The focus of the manuscript is assessing the role of CX32 in the urothelium and in luminal cancers.

The studies use 2 human cell models that appear to generate different "differentiated" phenotypes, based on the images in the manuscript. The undifferentiated culture appears to be squamous-like, while the differentiated cultures should be urothelial-like, however there are significant differences in size and morphology between the 2 types of differentiated cultures, suggesting that the populations may differ quite a bit. This is a huge problem, because we are modeling the urothelium, but we don't know what urothelial cell types are present/represented in the cultures, making the rest of the data less important and impossible to interpret. ABSCa (serum and physiological calcium) generates cells that look morphologically like superficial/facet cells-large and connected via CX32+ junctions. CX32+ cells in cultures with pharmacological activation of PPARG in EGFR-blocked cells using troglitazone and PD153035 (TZPD) are much smaller, with smaller nuclei. These cultures also contain unlabeled cells-presumably a different population (what are these?)that do not express much CX32. These differentiation protocols are used together in some cases, or ABSCa is used alone resulting more confusion and a general lack of consistency.

It appears that as stated early in the manuscript, differentiation is linked to changes in cell state-(this is not even mentioned) but what cell types are present in these 2 cultures? Are they representative of the normal bladder urothelium (it is stated that some are urethral urothelium in some cases-are these interchangeable with bladder urothelium?) Have these differentiated cultures been analyzed by sc-RNAseq to evaluate what which populations are present (especially important if as in Fig 1, cultures contain cells positive/negative for CX32)? Without a clear idea of which cell types are present before vs after, it is not possible to interpret the data in the paper.

Neither PPARG nor its targets are listed in the differentiation column in Fig 1, hence it is hard to know what role this pathway may play in the two conditions, nor whether expression or activity are similar to what is observed in vivo. That PPARG antagonist has little effect on dimer formation in ABSCa cultures, supporting the idea that these two models are quite different, and that we are looking at changes in populations that drive changes in gene expression and cell behavior. Based on what is known about top cells (superficial/facet cells) in the urothelium, they are post-mitotic and would probably not be able to undergo EMT. Then which cell types are undergoing EMT?

T134A mutations have not been characterized in the urothelium, and the effects of this mutation are not shown in the manuscript (that I could find). What is the expected phenotype?

Several of the experiments are not well designed. Experiments in Fig. 2 for example, were performed with WT cells (which are actually over expressing CX32) and mutants expressing T134A without over-expressing CX32. This is basically comparing cells

with increased CX32 with those with endogenous CX32 levels co-expressing the transfected mutant. Not very consistent.

A good place to start related to CX32 and cancer would be to have a look at the gene in bladder cancers from patients based on the large and accessible data sets in TCGA or other database. One would expect a strong association with luminal subtype (even could be looked at in tumor cell lines, which are pretty well characterized). Does this luminal expression include NMIBC? or it is only luminal MIBC?

In summary, since we don't know the composition of the cultures used in these experiments, it is hard to draw conclusions about the role of CX32 in the normal or diseased urothelium.

Reviewer #2 (Comments to the Authors (Required)):

The manuscript by J. Hinley et al. describes the studies of connexin 32 (Cx32) and its role in the terminal urothelial differentiation program. The paper is scientifically valid offering novel clues on molecular mechanisms of urothelial differentiation, providing new insights on urothelial cell physiology, and its putative role in urothelial carcinogenesis.

The experiments are well planned and executed showing that Cx32 is primarily expressed in vivo in basolateral borders of superficial urothelial cells. The loss of Cx32 induced a mesenchymal migratory switch in vitro. The validation studies on human bladder cancer samples including the TCGA cohort identify the potential role of the Cx32 regulatory mechanisms in urothelial carcinogenesis.

My specific comments are as follows:

1. The transcriptional mRNAseq analysis on NHU lines expressing Cx32wt and Cx32T134A mutant are superficial and requires a more in-depth approach. The analyses should be expanded to comprehensive basal and luminal markers used in luminal/basal classification complemented by the analysis of urothelial and mesenchymal regulons as well as EMT markers. These analyses should be amplified by the quantitative assessments of basal to luminal and EMT scores.
2. Similarly, the validation studies on mRNAseq TCGA cohort are superficial and not convincing.
 - a. The main idea recurring in all the experiments is the potential luminal to basal switch of the phenotype therefore it is surprising that the authors used the consensus classifications with six classes instead of MD Anderson-based luminal/basal classification.
 - b. I recommend that the author use the TCGA cohort subclassified into luminal and basal subtypes using the MD Anderson classification and identify the subsets of tumors in each molecular subtype which upregulate or downregulate Cx32, and its target genes identified in the NHU Cx32T134A mutant experiments. This should be correlated with the BLT and EMT scores as well as urothelial/mesenchymal regulons expression status providing the quantitative assessment of the effect of Cx32 overexpression or loss.

The analytical approach outlined above is based on the available data from the author's experiments and publicly available TCGA cohort. The results of these analyses will significantly amplify the information on the role of Cx32 in urothelial differentiation and carcinogenesis.

In summary, this is an important and in general well executed study concerning novel molecular mechanisms involved in terminal urothelial differentiation controlled by Cx32. In the opinion of this Reviewer, novel insights on molecular mechanisms of urothelial differentiation are essential to move our knowledge on urothelial carcinogenesis forward. The additional analytical plan based on the available mRNAseq data from the author's experiments and the TCGA cohort is intended to amplify the information on the role of Cx32 in urothelial differentiation and carcinogenesis. I am looking forward to seeing this manuscript enhanced by the analyses outlined above.

Reviewer #3 (Comments to the Authors (Required)):

Hinley et al. present novel findings that Cx32 is upregulated during urothelial differentiation, while its loss triggers a "wound-healing" program with TGF β signaling and mesenchymal features. Overall, the study is thorough and introduces an important concept that Cx32-mediated GJIC helps maintain urothelial homeostasis. Indeed, this is as far as I know the first systematic study of connexins in human urothelium, making the work significant.

The major drawback, in my opinion, is that the transcriptomic analysis and follow-up verification analysis are performed only by comparing Cx32 WT versus Cx32-T134A overexpressing cells. For the verification of some of the specific targets identified from this analysis, it would be pertinent to perform a complementary analysis of Cx32 knockdown vs control under differentiation. Cx32-T134A may have specific gain-of-function attributes and may affect other connexins that play a role in the urothelium. Specifically, the altered SMAD/TGF β signalling observed with T134A should be tested in their Cx32 shRNAi system. I suggest the authors also discuss potential alternative mechanisms (channel-independent roles of Cx32 or effects on other connexins). Indeed, this mutant localizes to the membrane, whereas a key point from the immunohistochemistry analysis is the presence of cytoplasmic Cx32. This key difference should be taken into consideration and discussed.

Also, while the tissue microarray data link Cx32 localization to luminal subtype markers and higher Ki67/vimentin, this is correlative. The authors should be cautious in claiming a causal role in cancer behavior without patient outcome data or functional assays in tumor cells. A survival analysis stratified by Cx32 localization (if data exist) could be insightful. Indeed, the term biomarker should be reconsidered: while Cx32 localization correlates with a tumor subset, more evidence (e.g. prognostic value) would be needed to call it a biomarker in clinical terms.

One point to clarify: the manuscript suggests PPAR γ drives Cx32 expression. Since PPAR γ itself enforces luminal differentiation, it would be good to discuss whether Cx32 is merely a downstream marker of this program or has an independent role.

Reviewer #1

The focus of the manuscript is assessing the role of CX32 in the urothelium and in luminal cancers.

The studies use 2 human cell models that appear to generate different "differentiated" phenotypes, based on the images in the manuscript. The undifferentiated culture appears to be squamous-like, while the differentiated cultures should be urothelial-like, however there are significant differences in size and morphology between the 2 types of differentiated cultures, suggesting that the populations may differ quite a bit. This is a huge problem, because we are modeling the urothelium, but we don't know what urothelial cell types are present/represented in the cultures, making the rest of the data less important and impossible to interpret. ABSCa (serum and physiological calcium) generates cells that look morphologically like superficial/facet cells-large and connected via CX32+ junctions. CX32+ cells in cultures with pharmacological activation of PPAR γ in EGFR-blocked cells using troglitazone and PD153035 (TZPD) are much smaller, with smaller nuclei. These cultures also contain unlabeled cells-presumably a different population (what are these?) that do not express much CX32. These differentiation protocols are used together in some cases, or ABSCa is used alone resulting more confusion and a general lack of consistency.

We would like to thank Reviewer 1 for their interest in our normal human urothelial *in vitro* cell culture system. This culture system has been robustly and extensively characterised in earlier publications (key references are referred to in the manuscript) and includes the capacity of NHU cells to acquire a more differentiated urothelial phenotype in response to extrinsic triggers such as exposure to serum or PPAR γ activation (Fleming et al., 2012).

As the reviewer indicates, the outcomes of different differentiation-inducing protocols are not identical. However, both protocols do enable differentiation to proceed, leading to the *de novo* expression of genes that characterise mature superficial urothelial cells *in situ*. Rather than making our data less important, we reasoned that inclusion of results based on both protocols added robustness to our study, by verifying that differentiation-associated *GJB1* gene expression is not a methodology artefact. It also enabled us to exploit inherent differences between the two differentiation models to our experimental advantage. For example:

- The SLDT assay (shown in Fig 2) required assessment of cell-cell communication by means of a gap junction permeable fluorescent dye. This was elegantly imaged and quantified using dye-transfer fluorescence microscopy in TZPD-differentiated cultures that remained minimally-stratified, whilst the ABSCa model was technically less tractable to this approach due to its stratified nature.
- The ABSCa protocol generates a barrier forming, multilayered epithelial sheet expressing markers associated with urothelial differentiation and representative of normal urothelium, in terms of organisation and expression of basal, intermediate and superficial cell markers. The barrier-forming properties of this model were essential to our assessment of barrier repair monitored by measuring Transepithelial Electrical Resistance (TEER), in Fig 3.

To meet the point raised by the reviewer that our manuscript would be improved by providing more context, including a description of the advantages and limitations for the two different modes of differentiating urothelial cells, we have included a new paragraph to the Introduction to describe the two differentiation models (highlighted text). We have otherwise been clear throughout the manuscript (and in figure captions) to specify which differentiation protocol has been used.

It appears that as stated early in the manuscript, differentiation is linked to changes in cell state-(this is not even mentioned) but what cell types are present in these 2 cultures? Are they representative of the normal bladder urothelium (it is stated that some are urethral urothelium in some cases-are these interchangeable with bladder urothelium?) Have these differentiated cultures been analyzed by sc-RNAseq to evaluate what which populations are present (especially important if as in Fig 1, cultures contain cells positive/negative for CX32)? Without a clear idea of which cell types are present before vs after, it is not possible to interpret the data in the paper.

The reviewer queries whether urethral and bladder-derived NHU cells are interchangeable. Urothelium is a transitional epithelium located along the urinary tract from renal pelvis, ureters, bladder, but excluding the urethra, which we do not study. By contrast, both bladder and ureter are lined by urothelium and though they differ in embryological origin, they are histologically comparable and indistinguishable in their expression profile of urothelial differentiation markers. We have explicitly used both bladder and ureteric-derived NHU cells in our study and show *GJB1* expression increased upon differentiation of NHU cells from both bladder and ureter.

We have not performed sc-RNAseq analysis in this study. Urothelium is constitutively heterogeneous as the cells adopt different phenotypes according to the different positions occupied – ie basal, intermediate and apical/superficial compartments. These positional phenotypes are interchangeable, as demonstrated by (Wezel et al., 2014), who showed reconstitution of complete urothelium from basal- or supra-basal derived urothelial cells. Where relevant, descriptions of the differentiation models used has been extended within our manuscript introduction. For clarity, we have now expanded our description of Cx32 by indirect immunofluorescence localisation to illustrate the expression by a subpopulation of cells in areas of stratification, (Figure 1).

Neither PPARG nor its targets are listed in the differentiation column in Fig 1, hence it is hard to know what role this pathway may play in the two conditions, nor whether expression or activity are similar to what is observed in vivo.

We have now incorporated *PPARG* expression into Figure 1A to demonstrate the positive correlation between differentiation and *PPARG* expression. Having said this, we would like to stress for the reviewers that it is not overall *PPARG* expression levels which drive urothelial differentiation, but ligand activation, so we have avoided overinterpretation of this in the manuscript. We haven't included data in the manuscript to compare *in vivo* expression to *in vitro* expression, since the objective was not to assess how closely the *in vitro* models recapitulated the *in vivo* scenario. Rather, we have made use of the cell culture

models available to us, as tools to study the development of gap junctions during the cytodifferentiation of NHU cells.

That PPARG antagonist has little effect on dimer formation in ABSCa cultures, supporting the idea that these two models are quite different, and that we are looking at changes in populations that drive changes in gene expression and cell behavior.

Our findings demonstrate that whilst PPARG activation (as used in the TZPD model) is one mode for stimulating Cx32 expression, there are other factors as present in serum able to drive Cx32 expression. In the results description for Figure 1D, we have already stated that additional drivers of Cx32 expression are likely present in serum.

Based on what is known about top cells (superficial/facet cells) in the urothelium, they are post-mitotic and would probably not be able to undergo EMT. Then which cell types are undergoing EMT?

We respectively point out to the reviewer that urothelium is a mitotically-quiescent epithelium but unlike other stratified epithelia that undergo irreversible loss of mitotic capacity during commitment to differentiation, all urothelial cells (including superficial) retain mitotic capacity. Our EMT signatures are based on bulk RNAseq of cell culture populations and therefore do not address this particular query. In Cx32 overexpressing (WT and T134A) cultures, we see an abundance of overexpressed Cx32 throughout the culture. We wouldn't expect this overexpression model to perfectly recapitulate native urothelium, but is an excellent tool for studying the consequence of ablating Cx32 function within differentiated urothelial cells.

T134A mutations have not been characterized in the urothelium, and the effects of this mutation are not shown in the manuscript (that I could find). What is the expected phenotype?

The original research paper describing this mutation provides some discussion as to why the T134A mutation has not been documented as a naturally occurring mutation. There are >250 distinct mutations in Cx32 creating manifestations of CMTX disease identified and published (Beahm et al., 2006), including the 4 upstream and 1 downstream amino acid residues identified as disease-causing. Those authors speculate that the most likely scenario is that the mutation would be embryonically lethal due to the almost complete knock-out of function in a dominant negative fashion which may cause "widespread effects on co-expressed connexins, eliminating any redundancy protections in tissues". The biological consequences of such a mutation are likely to be more severe.

Further to this, we do not suggest that this is a cancer driving mutation. We engineered genetically modified NHU cells with this mutation as a specific tool to examine the phenotypic consequence of blocking Cx32-mediated cell-cell communication.

Several of the experiments are not well designed. Experiments in Fig. 2 for example, were performed with WT cells (which are actually over expressing CX32) and mutants expressing T134A without over-expressing CX32. This is basically comparing cells with increased CX32

with those with endogenous CX32 levels co-expressing the transfected mutant. Not very consistent.

This comment suggests the reviewer has misunderstood the experimental system, which is fundamental to interpreting the manuscript. We have engineered cells with stable, transduced (rather than transfected) overexpression of Cx32T134A, containing a point mutation, which causes the hexameric channel to be locked in a permanent, dominant-negative closed state, even if some of the Cx32 subunits are wild-type. This generates a culture with an almost complete (dominant negative) knock-out of function, as described in the original paper detailing this mutation for all α and β -group connexins (Beahm et al., 2006). We have then compared this to Cx32 wild-type overexpressing cells, as an appropriately balanced comparison. Thus, we generated 2 cultures, one with an abundance of functional Cx32 and one with an abundance of non-functional Cx32, which we verified in our Scrape-load dye-transfer assays.

A good place to start related to CX32 and cancer would be to have a look at the gene in bladder cancers from patients based on the large and accessible data sets in TCGA or other database. One would expect a strong association with luminal subtype (even could be looked at in tumor cell lines, which are pretty well characterized). Does this luminal expression include NMIBC? or it is only luminal MIBC?

Thank you for this comment. We would like to alert the reviewer to our data in the manuscript which shows *GJB1* gene expression in the TCGA bladder cancer dataset, as the largest publicly available cohort of MIBCs. This can be found in our results section titled "Cx32 expression in luminal MIBC" and in Supplementary Figure 3. We have not included non-muscle invasive disease cohorts.

In summary, since we don't know the composition of the cultures used in these experiments, it is hard to draw conclusions about the role of CX32 in the normal or diseased urothelium.

Reviewer #2

The manuscript by J. Hinley et al. describes the studies of connexin 32 (Cx32) and its role in the terminal urothelial differentiation program. The paper is scientifically valid offering novel clues on molecular mechanisms of urothelial differentiation, providing new insights on urothelial cell physiology, and its putative role in urothelial carcinogenesis.

The experiments are well planned and executed showing that Cx32 is primarily expressed in vivo in basolateral borders of superficial urothelial cells. The loss of Cx32 induced a mesenchymal migratory switch in vitro. The validation studies on human bladder cancer samples including the TCGA cohort identify the potential role of the Cx32 regulatory mechanisms in urothelial carcinogenesis.

My specific comments are as follows:

1. The transcriptional mRNAseq analysis on NHU lines expressing Cx32wt and Cx32T134A

mutant are superficial and requires a more in-depth approach. The analyses should be expanded to comprehensive basal and luminal markers used in luminal/basal classification complemented by the analysis of urothelial and mesenchymal regulons as well as EMT markers. These analyses should be amplified by the quantitative assessments of basal to luminal and EMT scores.

We thank the reviewer for their constructive review and by the insightful suggestions.

Our data analyses revealed that the Cx32 loss of function mutant cells maintained their differentiated status whilst gaining markers of TGF β signalling and some EMT markers. However, we didn't see gain in classical "basal/squamous" markers. On the basis of the reviewer's suggestion, we have extended our supplementary panel (Figure S2A) to include expression of basal and luminal markers, based on the MD Anderson-based luminal/basal classification (Choi et al., 2014), with expansion to the results description for this section. This didn't change our overall finding that Cx32 pore-closed mutants maintained transitional (urothelial) differentiation attributes and did not gain basal/squamous features.

In Figure 4D, we provide a summary of EMT transcriptional regulators, as well as key EMT downstream targets as discussed in the results text under section "Transcriptomic analysis of Cx32-modified cells".

Whilst the basal to luminal switch is not relevant here, the reviewer suggests that we include a quantitative assessment EMT score. We have included (Figure 4B) our GSEA analyses which perform a quantitative assessment of our Cx32 mutant gene set against hallmark EMT genes. The normalised enrichment score was 2.181 (Cx32^{T134A} vs Cx32^{WT}) and P value 0.000 (to 3 decimal places). It's not clear from the comment if the reviewer is referring to an alternative comparison we could make to assess the EMT score, or whether our GSEA data has been missed during their review. Therefore, for clarity, we have updated the results text (in section "Transcriptomic analysis of Cx32-modified cells"), to specify the enrichment scores, for both Hallmark EMT, but also the Foroutan TGF β _EMT GSEA set.

2. Similarly, the validation studies on mRNAseq TCGA cohort are superficial and not convincing.

a. The main idea recurring in all the experiments is the potential luminal to basal switch of the phenotype therefore it is surprising that the authors used the consensus classifications with six classes instead of MD Anderson-based luminal/basal classification.

Our overarching finding is that Cx32 expression is a feature of a) differentiated (in normal) and b) luminal restricted biology (in MIBC). Given the reviewer's comment, we felt it would be important to make this point more pronounced in the paper, as we did not aim to suggest that Cx32 loss of function switches cells to a basal phenotype. Our point of interest is that loss of Cx32 communication drives expression of TGF β and EMT signalling on a differentiated transitional background, therefore its context is most relevant to luminal (MIBC) biology. Cx32 gene expression itself (being downstream of PPARG activity) is a feature of luminal biology and we wanted to make this point in a direct way that was

relevant to the manuscript. Our choice to use the consensus classifier was a) that it does take into account comprehensive basal/luminal markers in the classification whilst providing additional information on stromal contamination of the samples and b) it is the accepted consensus framework within the bladder cancer field.

Further to this, we have made some changes to both our abstract (final paragraph) and introduction (final paragraph) to further stress the point that our take home message is that Cx32 function/localisation within the context of differentiated/luminal biology is informative of invasive characteristics and reveals the potential for differentiated urothelial cancers to exhibit EMT.

b. I recommend that the author use the TCGA cohort subclassified into luminal and basal subtypes using the MD Anderson classification and identify the subsets of tumors in each molecular subtype which upregulate or downregulate Cx32, and its target genes identified in the NHU Cx32T134A mutant experiments. This should be correlated with the BLT and EMT scores as well as urothelial/mesenchymal regulons expression status providing the quantitative assessment of the effect of Cx32 overexpression or loss.

The analytical approach outlined above is based on the available data from the author's experiments and publicly available TCGA cohort. The results of these analyses will significantly amplify the information on the role of Cx32 in urothelial differentiation and carcinogenesis.

This misses our fundamental point that Cx32 is implicit to urothelial/luminal differentiation. As described in response to point 1 from this reviewer, we have taken the TCGA cohort and subclassified it already into basal/luminal subtypes, based on the consensus classifier, rather than MD Anderson, as reasoned for point 2a above (Supplementary Figure 3). As stated in the manuscript, Cx32 gene expression is associated with all luminal subtypes in the consensus classifier.

We did perform further analyses of TCGA to examine the regulon of genes upregulated in differentiated normal NHU cells with mutated (pore-closed) Cx32. As this is suggested by the reviewer, we enclose a figure (below) of this regulon mapped as a heatmap in the TCGA-luminal cohort (NB: we were interested to see if tumours could be split on the basis of dysfunctional Cx32, and therefore basal tumours, which lack Cx32 were excluded). The data reveal that:

- In the TCGA, our Cx32 mutant regulon is enriched in stroma rich and luminal Non-specified tumours.
- Outcomes are significantly worse in patients who have high expression of this regulon.

This has important implications as both stroma-rich and luminal NS groups have high stromal infiltrates. Whilst our regulon is based on differentiated urothelial biology, most of the markers in the regulon are stroma enriched (as EMT-like markers). Therefore, despite intriguing survival findings, we felt the conundrum of distinguishing between tumour EMT versus stromal contamination of tumour made it impossible for us to use the TCGA dataset to draw conclusions about “EMT-like” tumour cell/epithelial biology. Without being able to separate the potential influence of luminal EMT from stroma, we felt it was too controversial

to include this figure in the manuscript. We have already included a paragraph to describe this issue in our discussion (penultimate paragraph), but we elected not to show the data for these reasons. Access to a single cell tumour dataset would have been ideal, but we do not have such data available.

In summary, this is an important and in general well executed study concerning novel molecular mechanisms involved in terminal urothelial differentiation controlled by Cx32. In the opinion of this Reviewer, novel insights on molecular mechanisms of urothelial differentiation are essential to move our knowledge on urothelial carcinogenesis forward. The additional analytical plan based on the available mRNAseq data from the author's experiments and the TCGA cohort is intended to amplify the information on the role of Cx32 in urothelial differentiation and carcinogenesis. I am looking forward to seeing this manuscript enhanced by the analyses outlined above.

Reviewer #3

Hinley et al. present novel findings that Cx32 is upregulated during urothelial differentiation, while its loss triggers a "wound-healing" program with TGF β signaling and mesenchymal features. Overall, the study is thorough and introduces an important concept that Cx32-mediated GJIC helps maintain urothelial homeostasis. Indeed, this is as far as I know the first systematic study of connexins in human urothelium, making the work significant.

We would like to thank the reviewer for their valuable and constructive feedback.

The major drawback, in my opinion, is that the transcriptomic analysis and follow-up verification analysis are performed only by comparing Cx32 WT versus Cx32-T134A overexpressing cells. For the verification of some of the specific targets identified from this analysis, it would be pertinent to perform a complementary analysis of Cx32 knockdown vs control under differentiation. Cx32-T134A may have specific gain-of-function attributes and may affect other connexins that play a role in the urothelium. Specifically, the altered SMAD/TGF β signalling observed with T134A should be tested in their Cx32 shRNAi system.

The reviewer makes a very valid point. In our research we generated both Cx32 shRNA knock-down cells, as well as the dominant-negative Cx32T134A overexpression system (and parallel control Cx32WT cells). Our rationale for prioritising the overexpression system for transcriptomic analysis (rather than shRNA) was for several reasons:

- 1) We didn't achieve full knock-down. In testing of three independent GJB1 shRNA sequences in our retroviral vectors, we achieved ~50% reduction in protein expression.
- 2) We observed similar findings with the Cx32 shRNA knock-down versus the Cx32T134A dominant negative in terms of migratory changes and a gain of vimentin and pSMAD protein expression. However, we rationalised that there was an advantage to studying the overexpression of Cx32WT as an additional "gain of function" control, since the overall Cx32 protein pool and protein interactions should be in keeping with the Cx32T134A arm.
- 3) In other cell systems, connexins are reported to have protein interactions, for example within tight junctions, as well as cell signalling roles. Knocking down Cx32 could pose a

drawback in our experimental system, since destabilising protein interactions could have unintended consequences. We therefore took a preference for genetically modifying cells in a way which allowed for connexin expression and protein interactions between Cx32T134A and Cx32WT to be balanced, allowing us to specify effects which were communication (functional) rather than structurally dependent.

Nonetheless, as the reviewer recommends, we do have preliminary data which suggests that we see similar effects on Vimentin, pSMAD3 and pSMAD1/5 when we knock down Cx32 using our shRNA system. We are happy to include this extra immunoblotting, to complement the more detailed transcriptomic analysis and follow-up protein expression data performed on the Cx32T134A overexpressing cells. We have included an extra 2 panels (Supplementary Fig 2, H & I) and a description of the findings in Results section “The altered phenotype of Cx32-modified cells” to display these findings.

I suggest the authors also discuss potential alternative mechanisms (channel-independent roles of Cx32 or effects on other connexins). Indeed, this mutant localizes to the membrane, whereas a key point from the immunohistochemistry analysis is the presence of cytoplasmic Cx32. This key difference should be taken into consideration and discussed.

Thank you for this comment. We have added two paragraphs to our discussion which take into account the possibility of:

- a) channel-independent effects of overexpressing Cx32, such as accumulation of cytoplasmic Cx32
- b) The potential for effects of Cx32 overexpression on other connexins.

Also, while the tissue microarray data link Cx32 localization to luminal subtype markers and higher Ki67/vimentin, this is correlative. The authors should be cautious in claiming a causal role in cancer behavior without patient outcome data or functional assays in tumor cells. A survival analysis stratified by Cx32 localization (if data exist) could be insightful. Indeed, the term biomarker should be reconsidered: while Cx32 localization correlates with a tumor subset, more evidence (e.g. prognostic value) would be needed to call it a biomarker in clinical terms.

This is a fair comment and we have made some changes to the final paragraph of the abstract, and final paragraph of the discussion to avoid overstating.

One point to clarify: the manuscript suggests PPAR γ drives Cx32 expression. Since PPAR γ itself enforces luminal differentiation, it would be good to discuss whether Cx32 is merely a downstream marker of this program or has an independent role.

We have included a sentence in the discussion to clarify that the relationship between PPAR γ and Cx32 is not necessarily direct or causal, but their biology is intrinsically linked.

- BEAHM, D. L., OSHIMA, A., GAIETTA, G. M., HAND, G. M., SMOCK, A. E., ZUCKER, S. N., TOLOUE, M. M., CHANDRASEKHAR, A., NICHOLSON, B. J. & SOSINSKY, G. E. 2006. Mutation of a conserved threonine in the third transmembrane helix of alpha- and beta-connexins creates a dominant-negative closed gap junction channel. *J Biol Chem*, 281, 7994-8009.
- CHOI, W., PORTEN, S., KIM, S., WILLIS, D., PLIMACK, E. R., HOFFMANN-CENSITS, J., ROTH, B., CHENG, T., TRAN, M., LEE, I. L., MELQUIST, J., BONDARUK, J., MAJEWSKI, T., ZHANG, S., PRETZSCH, S., BAGGERLY, K., SIEFKER-RADTKE, A., CZERNIAK, B., DINNEY, C. P. & MCCONKEY, D. J. 2014. Identification of distinct basal and luminal subtypes of muscle-invasive bladder cancer with different sensitivities to frontline chemotherapy. *Cancer Cell*, 25, 152-65.
- FLEMING, J. M., SHABIR, S., VARLEY, C. L., KIRKWOOD, L. A., WHITE, A., HOLDER, J., TREJDOSIEWICZ, L. K. & SOUTHGATE, J. 2012. Differentiation-associated reprogramming of the transforming growth factor beta receptor pathway establishes the circuitry for epithelial autocrine/paracrine repair. *PLoS One*, 7, e51404.
- WEZEL, F., PEARSON, J. & SOUTHGATE, J. 2014. Plasticity of in vitro-generated urothelial cells for functional tissue formation. *Tissue Eng Part A*, 20, 1358-68.

In differentiated NHU cells, Connexin 32 T134A induced 49 genes significantly >2-fold. To evaluate whether this Cx32 regulon was detectable in tumours, TCGA_BLCA expression data was downloaded from the National Cancer Institute (NCI) Genomic Data Commons dbGaP accession phs000178.v10.p8 as part of dbGaP project 19625. Counts data were converted to Transcripts Per Million (TPM) values.

Spearman rank correlation matrices were used to evaluate co-regulation of the regulon within luminal tumours. Since the "Stroma-Rich" class of the consensus molecular classification describes urothelial purity of the sample (ie the presence of other cell types) and not urothelial differentiation as for the remaining 5 classes, the published centroids were used to reclassify the Stroma-rich class as the best fit urothelial class (ie Ba/Sq, LumNS, LumP, LumU and NE-like).

Genes have multiple modes of regulation in situ that may not have been present in vitro and so in order to retain specificity in the Cx32 regulon, genes with divergent in situ expression patterns were removed (A, in grey). The Spearman rank correlation matrices revealed a core groups of 25 genes (A, in red) where the median Spearman Rho for comparison with the rest of the Connexin 32 DN induced genes in TCGA luminal tumours was significant ($p < 0.05$; i.e. the gene was significantly correlated with >50% of the Connexin 32 DN induced genes). This process removes genes that were regulated by Cx32 disruption in vitro but were regulated more strongly by alternative mechanisms in situ.

The Cx32 regulon was enriched in LumNS tumours of TCGA cohort and the luminal tumours form two groups using Euclidean distance to perform K means clustering into two groups (B) with a significant difference in survival (C) between the two groups.

January 9, 2026

RE: Life Science Alliance Manuscript #LSA-2025-03427-TR

Prof. Jennifer Southgate
University of York
Department of Biology
Jack Birch Unit
Heslington
York YO10 5DD
United Kingdom

Dear Dr. Southgate,

Thank you for submitting your revised manuscript entitled "Connexin 32 constrains a mesenchymal-like switch in differentiated urothelium and luminal cancers". We apologise for the delay in communicating our decision due to editor availability issues, and in securing and clarifying reviewer comments.

Your manuscript was reviewed by two of the original reviewers. Reviewer 3 has acknowledged that your revisions have strengthened the manuscript. You will note that Reviewer 2 had pending reservations regarding the resolution of their original concerns in the revised manuscript. We communicated with this reviewer who kindly clarified their views. In line with their comments on your revised manuscript, we recommend that you include the following (1) a rationale in the manuscript text for not using the MD-Anderson classifier, as you have done in your response letter, and (2) analyses of TCGA along with related descriptions in the main manuscript text as already provided in Appendix 2 of your response letter. We recognise that you have a reservation to draw conclusions on "EMT-like" tumour cell/epithelial biology based on the figure in Appendix 2 (in the response letter). Nevertheless we strongly encourage you to include this as a supplementary figure whilst you allude to the potential stromal contamination (with this experiment) in line 477 of the discussion.

Based on the overall reviewers' evaluation, we would be happy to publish your paper in Life Science Alliance pending resolution of the above requests, and final revisions necessary to meet our formatting guidelines.

-Please confirm if the following images are the same and if so, please indicate this in both the respective figure legends:

(a) 6A (membrane and cytoplasmic) AND 6F (membrane and cytoplasmic, under Cx32)

(b) part of images in 6A in row titled, Negative, for GATA3 AND FOXA1.

-We thank you for disclosing the Department of Biology Ethics Committee as the research approving committee. We recommend that you also include the organisation/university this department is affiliated with. Please also include a statement if informed consent was obtained from all subjects and that the experiments conformed to the principles set out in the WMA Declaration of Helsinki and the Department of Health and Human Services Belmont Report (as suggested in LSA's guidelines).

-We recommend that you provide a citation for Promoter analyses or provide full details on that method.

-For description of microscopy, please provide details for objectives used (type, name, NA). For timelapse imaging, please include temperature of image acquisition.

-Please add molecular weight markers to the images of blots in Figure 2A.

-Thank you for providing the underlying source data. We recommend that you refer to source data files in the manuscript.

-Please add ORCID ID for secondary corresponding author - they should have received instructions on how to do so.

-Please add the X and Bluesky handles of your host institute/organisation, as well as your own and/or one of the authors, in our system.

-The "Data Availability" section should be placed after the Materials & Methods section. Please consult our guidelines at <https://www.life-science-alliance.org/manuscript-prep#format>

-Please add an Author Contributions section to your main manuscript text.

-Please add your main, supplementary figure, and table legends to the main manuscript text after the references section.

-It is recommended to exclude figures from the manuscript text and upload them separately.

-We encourage you to revise the figure legends for Figure S3 such that the figure panels match the actual figure.

-Please use the [10 author names et al.] format in your references (i.e., limit the author names to the first 10).

-Please add callouts for Figures 6F; S3C and S4A-D to your main manuscript text.

-We recommend that you rename section titled "Author Declaration" as "Conflicts of Interest Statement".

-Please be sure that the authorship listing and order is correct.

LSA now encourages authors to provide a 30-60 second video where the study is briefly explained. We will use these videos on

social media to promote the published paper and the presenting author (for examples, see <https://docs.google.com/document/d/1-UWCfbE4pGcDdcgzcmiuJl2XMBJnxKYeqRvLLrLSo8s/edit?usp=sharing>). Corresponding or first-authors are welcome to submit the video. Please submit only one video per manuscript. The video can be emailed to contact@life-science-alliance.org

A. FINAL FILES:

B. MANUSCRIPT ORGANIZATION AND FORMATTING:

Thank you for your attention to these final processing requirements. Please revise and format the manuscript and upload materials as soon as you are able.

Sincerely,

Sarita Hebbar, PhD
Scientific Editor
Life Science Alliance
<http://www.lsajournal.org>

Reviewer #2 (Comments to the Authors (Required)):

Unfortunately, the authors did not address positively not even one of my original comments. The comments from my original

review are still valid and remain not adequately addressed.

The recommended re-analyses from my original review are easy to accomplish and they are numerous published examples on how to address them. The recommended references to help the authors to re-analyze the data are as follows:

1. C.C. Guo et al. Sci Rep. 2020 Jun 16;10(1):9743. doi: 10.1038/s41598-020-66747-7. PMID: 32546765; PMCID: PMC7298008.

2. L. A. Byers, et al. Clin Cancer Res. 2013 Jan 1;19(1):279-90. doi: 10.1158/1078-0432.CCR-12-1558. Epub 2012 Oct 22. PMID: 23091115; PMCID: PMC3567921.

Reviewer #3 (Comments to the Authors (Required)):

Although further analysis of Cx32 knockdown or (ideally) KO conditions would have strengthened the data, overall the manuscript provides valuable insights, and I support its publication.

Dear Dr Hebbbar,

Thank you for your constructive comments and agreement to publish our manuscript "Connexin 32 constrains a mesenchymal-like switch in differentiated urothelium and luminal cancers".

We have made changes to the manuscript (which are highlighted) to address the reviewers points as described below.

We hope this meets the requirements now for publication, but please don't hesitate to contact us if you have any further requests.

Yours sincerely

Jenny Southgate & Jenny Hinley

Reviewer/Editor Points to address

(1) include rationale in the manuscript text for not using the MD-Anderson classifier.

Included in the penultimate paragraph of the Discussion and by changes to the Results - both changes are highlighted in the edited version of the revised manuscript.

(2) include analyses of TCGA along with related descriptions in the main manuscript text as already provided in Appendix 2 of your response letter.

We have included this data as suggested, with relevant text added to the Methods, the final paragraph of the Results and penultimate paragraph of the Discussion, along with a new supplementary Figure 4 and caption. Changes are highlighted in the edited version of the revised manuscript.

In addition, we have attended to the Editorial requests as below.

Please confirm if the following images are the same and if so, please indicate this in both the respective figure legends:

- (a) 6A (membrane and cytoplasmic) AND 6F (membrane and cytoplasmic, under Cx32)**
- (b) part of images in 6A in row titled, Negative, for GATA3 AND FOXA1.**

We apologise for any confusion over Figure 6 and have revised Figure legends to make clear which tumour biopsies are referred to.

Figure 6A relates to three individual tumours shown in top, middle and lower panels, illustrating tumours where Cx32 is localised as membrane, cytoplasmic or absent, respectively. To the right, these panels further show the relative status of GATA3 and FOXA1 as luminal markers in these tumours. Serial sections have been used to localise different antibodies to the same tumour region.

Figure 6F shows the same tumours as shown in 6A that display membrane- or cytoplasmic-localised Cx32. This is now clarified in the caption. This is highlighted in the edited version.

We thank you for disclosing the Department of Biology Ethics Committee as the research approving committee. We recommend that you also include the organisation/university this department is affiliated with. Please also include a statement if informed consent was obtained from all subjects and that the experiments conformed to the principles set out in the WMA Declaration of Helsinki and the Department of Health and Human Services Belmont Report (as suggested in LSA's guidelines).

We have expanded our section in the Materials & Methods titled “Tissue collection”, to address these points; this is highlighted in the edited manuscript version.

We recommend that you provide a citation for Promoter analyses or provide full details on that method.

Reference and website have been updated in Materials & Methods section for “Promoter Analysis”.

For description of microscopy, please provide details for objectives used (type, name, NA).

Included

For timelapse imaging, please include temperature of image acquisition.

Included

Please add molecular weight markers to the images of blots in Figure 2A.

Included

Thank you for providing the underlying source data. We recommend that you refer to source data files in the manuscript.

We have referred to this under the method for immunoblotting (highlighted).

Please add ORCID ID for secondary corresponding author - they should have received instructions on how to do so.

The Orcid ID for Jennifer Hinley (ID 0000-0001-8128-6669) has been linked. This has been confirmed by email on 4th Dec 2025 (by Jelena Kohajm, Editorial Assistant @ LSA) that my ORCID iD is now linked with the LSA account.

Please add the X and Bluesky handles of your host institute/organisation, as well as your own and/or one of the authors, in our system.

X accounts

Host institution X accounts are as follows. Our authors no longer have X accounts.

@UniOfYork

@YBRI_UoY

@BiologyatYork

Bluesky

Host institution/author Bluesky accounts are as follows:

@biologyatyork.bsky.social

@ybri-uoy.bsky.social

@jennybaker.bsky.social

@drsimonbaker.bsky.social

@asmasonomics.bsky.social

Social media summary: “Hinley and colleagues reveal Connexin 32-mediated cell-cell communications maintain mitotic-quiescence and suppress migration-associated TGF β signalling, ECM remodelling and mesenchymal-like changes in human urothelium. These features establish a dichotomy in the biology of luminal muscle invasive bladder cancers”.

The "Data Availability" section should be placed after the Materials & Methods section. Please consult our guidelines at <https://www.life-science-alliance.org/manuscript-prep#format>

This has been included

Please add an Author Contributions section to your main manuscript text.

This has now been included, we have also included a conflict of interest statement.

Please add your main, supplementary figure, and table legends to the main manuscript text after the references section.

This has been addressed and the legends are now below the references.

It is recommended to exclude figures from the manuscript text and upload them separately.

They have been removed from the main text

We encourage you to revise the figure legends for Figure S3 such that the figure panels match the actual figure.

This has been rectified

Please use the [10 author names et al.] format in your references (i.e., limit the author names to the first 10).

This referencing change has been made.

Please add callouts for Figures 6F; S3C and S4A-D to your main manuscript text.

The callout for 6F is now inserted.

There is no S3C.

S4 A-D has now become S5A-D with the new supplementary figure inclusion based on the reviewer request. S5 is a methods figure and relates to the “Immunohistochemistry and semi-quantitative analysis” methods section, where there is already a call out for supplementary figure 5.

We recommend that you rename section titled "Author Declaration" as "Conflicts of Interest Statement".

A conflict of interest statement has been added

Please be sure that the authorship listing and order is correct.

We have checked this and all is correct.

January 30, 2026

RE: Life Science Alliance Manuscript #LSA-2025-03427-TRR

Prof. Jennifer Southgate
University of York
Jack Birch Unit, York Biomedical Research Institute and Department of Biology
Heslington
York YO10 5DD
United Kingdom

Dear Dr. Southgate,

Thank you for submitting your Research Article entitled "Connexin 32 constrains a mesenchymal-like switch in differentiated urothelium and luminal cancers". It is a pleasure to let you know that your manuscript is now accepted for publication in Life Science Alliance. Congratulations on this interesting work.

Your manuscript will now progress through copyediting and proofing. At the proofs stage, kindly (1) include details for objectives (type, magnification, NA) in the description of timelapse imaging, (2) rephrase the legend for Figure 6A to indicate use of serial sections for each tumour (top, middle, lower) for GATA3/FOXA1 (as done in your response letter).

It is journal policy that authors provide original data upon request.

DISTRIBUTION OF MATERIALS:

Again, congratulations on a very nice paper. I hope you found the review process to be constructive and are pleased with how the manuscript was handled editorially. We look forward to future exciting submissions from your lab.

Sincerely,

Sarita Hebbbar, PhD
Scientific Editor
Life Science Alliance
<http://www.lsajournal.org>